# Uncertain Decisions Facilitate Better Preference Learning

**Cassidy Laidlaw**
University of California, Berkeley
cassidy_laidlaw@cs.berkeley.edu

**Stuart Russell**
University of California, Berkeley
russell@cs.berkeley.edu

## Abstract

Existing observational approaches for learning human preferences, such as inverse reinforcement learning, usually make strong assumptions about the observability of the human's environment. However, in reality, people make many important decisions under uncertainty. To better understand preference learning in these cases, we study the setting of inverse decision theory (IDT), a previously proposed framework where a human is observed making non-sequential binary decisions under uncertainty. In IDT, the human's preferences are conveyed through their loss function, which expresses a tradeoff between different types of mistakes. We give the first statistical analysis of IDT, providing conditions necessary to identify these preferences and characterizing the sample complexity—the number of decisions that must be observed to learn the tradeoff the human is making to a desired precision. Interestingly, we show that it is actually easier to identify preferences when the decision problem is more uncertain. Furthermore, uncertain decision problems allow us to relax the unrealistic assumption that the human is an optimal decision maker but still identify their exact preferences; we give sample complexities in this suboptimal case as well. Our analysis contradicts the intuition that partial observability should make preference learning more difficult. It also provides a first step towards understanding and improving preference learning methods for uncertain and suboptimal humans.

## 1 Introduction

The problem of inferring human preferences has been studied for decades in fields such as inverse reinforcement learning (IRL), preference elicitation, and active learning. However, there are still several shortcomings in existing methods for preference learning. Active learning methods require query access to a human; this is infeasible in many purely observational settings and may lead to inaccuracies due to the description-experience gap [1]. IRL is an alternative preference learning tool which requires only observations of human behavior. However, IRL suffers from underspecification, i.e. preferences are not precisely identifiable from observed behavior [2]. Furthermore, nearly all IRL methods require that the observed human is optimal or noisily optimal at optimizing for their preferences. However, humans are often *systematically* suboptimal decision makers [3], and accounting for this makes IRL even *more* underspecified, since it is hard to tell suboptimal behavior for one set of preferences apart from optimal behavior for another set of preferences [4].

IRL and preference learning from observational data are generally applied in situations where a human is acting under no uncertainty. Given the underspecification challenge, one might expect that adding in the possibility of uncertainty in decision making (known as partial observability) would only make preference learning more challenging. Indeed, Choi and Kim [5] and Chinaei and Chaib-Draa [6], who worked to apply IRL to partially observable Markov decision processes (POMDPs, where agents

35th Conference on Neural Information Processing Systems (NeurIPS 2021).

|  | **Decisions without uncertainty** | **Decisions under uncertainty** |
|---|---|---|
| (a) | Should I quarantine a traveler with a 100% accurate negative test for a dangerous disease? | Should I quarantine a traveler with some symptoms of a dangerous disease but no test results? |
| (b) | Should a person with irrefutable evidence of and confession to a crime be convicted? | Should a person with circumstantial evidence of a crime be convicted? |

Figure 1: One of our key findings is that *decisions made under uncertainty can reveal more preferences than clear decisions.* Here we give examples of decisions made with and without uncertainty. (a) In the case without uncertainty, nobody would choose to quarantine the traveler, so we cannot distinguish between different people's preferences. However, in the case *with* uncertainty, people might decide differently whether to quarantine the traveler depending on their preferences on the tradeoff between individual freedom and public health. This allows us to identify those preferences by observing decisions. (b) Similarly, observing decisions on whether to convict a person under uncertainty reveals preferences about the tradeoff between convicting innocent people and allowing criminals to go free.

act under uncertainty), remarked that the underspecification of IRL combined with the intractability of POMDPs made for a very difficult task.

In this work, we find that, surprisingly, observing humans making decisions under uncertainty actually makes preference learning *easier* (see Figure 1). To show this, we analyze a simple setting, where a human decision maker observes some information and must make a binary choice. This is somewhat analogous to supervised learning, where a decision rule is chosen to minimize some loss function over a data distribution. In our formulation, the goal is to learn the human decision maker's loss function by observing their decisions. Often, in supervised learning, the loss function is simply the 0-1 loss. However, humans may incorporate many other factors into their implicit "loss functions"; they may weight different types of mistakes unequally or incorporate fairness constraints, for instance. One might call this setting "inverse supervised learning," but it is better described as inverse decision theory (IDT) [7, 8], since the objective is to reverse-engineer only the human's decision rule and not any learning process used to arrive at it. IDT can be shown to be a special case of partially observable IRL (see Appendix B) but its restricted assumptions allow more analysis than would be possible for IRL in arbitrary POMDPs. However, we believe that the insights we gain from studying IDT should be applicable to POMDPs and uncertain decision making settings in general. We introduce a formal description of IDT in Section 3.

While we hope to provide insight into general reward learning, IDT is also a useful tool in its own right; even in this binary, non-sequential setting, human decisions can reveal important preferences. For example, during a deadly disease outbreak, a government might pass a law to quarantine individuals with a chance of being sick. The decision rule the government uses to choose who to quarantine depends on the relative costs of failing to quarantine a sick person versus accidentally quarantining an uninfected one. In this way, even human decisions where there is a "right" answer are revealing if they are made under uncertainty. This example could distinguish a preference for saving lives versus one for guaranteeing freedom of movement. These preferences on the tradeoff between costs of mistakes are expressed through the loss function that the decision maker optimizes.

In our main results on IDT in Section 4, we find that the identifiability of a human's loss function is dependent on whether the decision we observe them making involves uncertainty. If the human faces sufficient uncertainty, we give tight sample complexity bounds on the number of decisions we must observe to identify their loss function, and thus preferences, to any desired precision (Theorem 4.2). On the other hand, if there is no uncertainty—i.e., the correct decision is always obvious—then we show that there is no way to identify the loss function (Theorem 4.11 and Corollary 4.12). Technically, we show that learning the loss function is equivalent to identifying a threshold function over the space of posterior probabilities for which decision is correct given an observation (Figure 2). This threshold can be determined to precision $\epsilon$ in $\Theta(1/(p_c\epsilon))$ samples, where $p_c$ is the probability density of posterior probabilities around the threshold. In the case where there is no uncertainty in the decision problem, $p_c = 0$ and we demonstrate that the loss function cannot be identified.

These results apply to optimal human decision makers—that is, those who completely minimize their expected loss. When a decision rule or policy is suboptimal, in general their loss function cannot be

learned [4, 9]. However, we show that decisions made under uncertainty are also helpful in this case; under certain models of suboptimality, we can still *exactly* recover the human's loss function.

We present two such models of suboptimality (see Figure 3). In both, we assume that the decision maker is restricting themselves to choosing a decision rule $h$ in some hypothesis class $\mathcal{H}$, which may not include the optimal decision rule. This framework is similar to that of agnostic supervised learning [10, 11], but solves the inverse problem of determining the loss function given a hypothesis class and decision samples. If the restricted hypothesis class $\mathcal{H}$ is known, we show that the loss function can be learned similarly to the optimal case (Theorem 4.7). Our analysis makes a novel connection between Bayesian posterior probabilities and binary hypothesis classes. However, assuming that $\mathcal{H}$ is known is a strong assumption; for instance, we might suspect that a decision maker is ignoring some data features but we may not know exactly which features. We formalize this case by assuming that the decision maker could be considering the optimal decision rule in any of a number of hypothesis classes in some family $\mathbb{H}$. This case is more challenging because we may need to identify which hypothesis class the human is using in order to identify their loss function. We show that, assuming a smoothness condition on $\mathbb{H}$, we can still obtain the decision maker's loss function (Theorem 4.10).

We conclude with a discussion of our results and their implications in Section 5. We extend IDT to more complex loss functions that can depend on certain attributes of the data in addition to the chosen decision; we show that this extension can be used to test for the fairness of a decision rule under certain criteria which were previously difficult to measure. We also compare the implications of IDT for preference learning in uncertain versus clear decision problems. Our work shows that uncertainty is *helpful* for preference learning and suggests how to exploit this fact.

## 2   Related Work

Our work builds upon that of Davies [8] and Swartz et al. [7], who first introduced inverse decision theory. They describe how to apply IDT to settings in which a doctor makes treatment decisions based on a few binary test outcomes, but provide no statistical analysis. In contrast, we explore when IDT can be expected to succeed in more general cases and how many observed decisions are necessary to infer the loss function. We also analyze cases where the decision maker is suboptimal for their loss function, which are not considered by Davies or Swartz et al.

Inverse reinforcement learning (IRL) [2, 12, 13, 14, 15], also known as inverse optimal control, aims to infer the reward function for an agent acting in a Markov decision process (MDP). Our formulation of IDT can be considered as a special case of IRL in a partially observable MDP (POMDP) with two states and two actions (see Appendix B). Some prior work explored IRL in POMDPs [5, 6] by reducing the POMDP to a belief-state MDP and applying standard IRL algorithms. Our main purpose is not to present improvements to IRL algorithms; rather, we give an analysis of the difference between observable and partially observable settings for preference learning. We begin with the restricted setting of IDT but hope to extend to sequential decision making in the future. We also consider cases where the human decision maker is suboptimal, which previous work did not explore.

Performance metric elicitation (ME) aims to learn a loss function (aka performance metric) by querying a human [16, 17, 18]. ME and other active learning approaches [19, 20, 21, 22] require the ability to actively ask a user for their preference among different loss or reward functions. In contrast, IDT aims to learn the loss function purely by observing a decision maker. Active learning is valuable for some applications, but there are many cases where it is infeasible. Observed decisions are often easier to obtain than expert feedback. Also, active learning may suffer from the description-experience gap [1]; that is, it may be difficult to evaluate in the abstract the comparisons that these methods give as queries to the user, leading to biased results. In contrast, observing human decision making "in the wild" with IDT could lead to a more accurate understanding of human preferences.

Preference and risk elicitation aim to identify people's preferences between different uncertain or certain choices. A common tool is to ask a person to choose between a lottery (i.e., uncertain payoff) and a guaranteed payoff, or between two lotteries, varying parameters and observing the resulting choices [23, 24, 25]. In our analysis of IDT, decision making under uncertainty can be cast as a natural series of choices between lotteries. If we observe enough different lotteries, the decision maker's preferences can be identified. On the other hand, if there is no uncertainty, then we only observe choices between guaranteed payoffs and there is little information to characterize preferences.

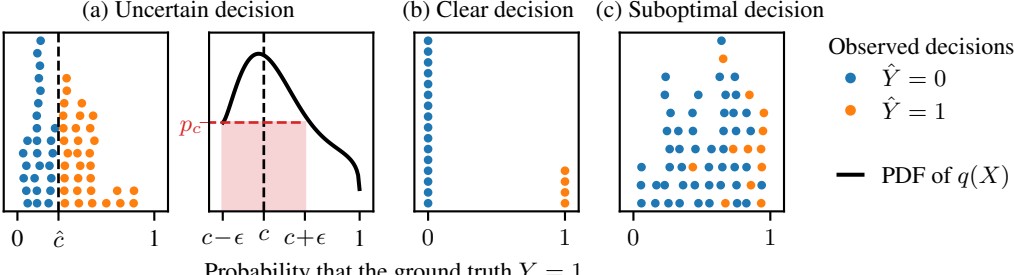

Figure 2: A visualization of three settings for inverse decision theory (IDT), which aims to estimate $c$, the parameter of a decision maker's loss function, given observed decisions $\hat{y}_1, \ldots, \hat{y}_m \in \{0, 1\}$. Here, each decision $\hat{y}_i$ is plotted against the probability $q(x_i) = \mathbb{P}(Y = 1 \mid X = x_i)$ that the ground truth (correct) decision $Y$ is 1 given the decision maker's observation $x_i$. Lemma 4.1 shows that an optimal decision rule assigns $\hat{y}_i = \mathbb{1}\{q(x_i) \geq c\}$. (a) For uncertain decision problems, IDT can estimate $c$ as the threshold of posterior probabilities $q(x_i)$ where the decision switches from 0 to 1 (Section 4.1). If the distribution of $q(X)$ has probability density at least $p_c$ on $[c - \epsilon, c + \epsilon]$, Theorem 4.2 shows we can learn $c$ to precision $\epsilon$ with $m \geq O(1/(p_c \epsilon))$ samples. (b) When there is no uncertainty in the decision problem, IDT cannot characterize the loss parameter $c$ because the threshold between positive and negative decisions could be anywhere between 0 and 1 (Section 4.4). (c) A suboptimal human decision maker does not use an optimal decision rule for any loss parameter $c$, but we can often still estimate their preferences (Sections 4.2 and 4.3).

## 3    Problem Formulation

We formalize inverse decision theory using decision theory and statistical learning theory. Let $\mathcal{D}$ be a distribution over observations $X \in \mathcal{X}$ and ground truth decisions $Y \in \{0, 1\}$. We consider an agent that receives an observation $X$ and must make a binary decision $\hat{Y} \in \{0, 1\}$. While many decision problems include more than two choices, we consider the binary case to simplify analysis. However, the results are applicable to decisions with larger numbers of choices; assuming irrelevance from independent alternatives (i.e. the independence axiom [26]), a decision among many choices can be reduced to binary choices between pairs of them. We generally assume that $\mathcal{D}$ is fixed and known to both the decision maker and the IDT algorithm. Unless otherwise stated, all expectations and probabilities on $X$ and $Y$ are with respect to the distribution $\mathcal{D}$.

We furthermore assume that the agent has chosen a decision rule (or hypothesis) $h : \mathcal{X} \to \{0, 1\}$ from some hypothesis class $\mathcal{H}$ that minimizes a loss function which depends only on the decision $\hat{Y} = h(X)$ that was made and the correct decision $Y$:

$$h \in \arg\min_{h \in \mathcal{H}} \; \mathbb{E}_{(X,Y) \sim \mathcal{D}} \left[ \ell(h(X), Y) \right].$$

In general, the loss function $\ell$ might depend on the observation $X$ as well; we explore this extension in the context of fair decision making in Section 5.1. Assuming the formulation above, since $Y, \hat{Y} \in \{0, 1\}$ we can write the loss function $\ell$ as a matrix $C \in \mathbb{R}^{2 \times 2}$ such that $\ell(\hat{y}, y) = C_{\hat{y}y}$. We denote by $\mathcal{R}_C(h) = \mathbb{E}_{(X,Y) \sim \mathcal{D}} \left[ \ell(h(X), Y) \right]$ the expected loss or "risk" of the hypothesis $h$ with cost matrix $C$. This cost matrix has four entries, but the following lemma shows that it effectively has only one degree of freedom.

**Lemma 3.1 (Equivalence of cost matrices).** *Any cost matrix* $C = \left( \begin{smallmatrix} C_{00} & C_{01} \\ C_{10} & C_{11} \end{smallmatrix} \right)$ *is equivalent to a cost matrix* $C' = \left( \begin{smallmatrix} 0 & 1-c \\ c & 0 \end{smallmatrix} \right)$ *where* $c = \frac{C_{10} - C_{00}}{C_{10} + C_{01} - C_{00} - C_{11}}$ *as long as* $C_{10} + C_{01} - C_{00} - C_{11} \neq 0$. *That is, there are constants* $a, b \in \mathbb{R}$ *such that* $\mathcal{R}_C(h) = a\mathcal{R}_{C'}(h) + b$ *for all h.*

See Appendix A.1 for this and other proofs. Based on Lemma 3.1, from now on, we assume the cost matrix only has one parameter $c$, which is the cost of a false positive; $1 - c$ is the cost of a false negative. Intuitively, high values of $c$ indicate a preference for erring towards the decision $\hat{Y} = 0$ under uncertainty while low values indicate a preference for erring towards the decision $\hat{Y} = 1$. Finally, we assume that making the correct decision is always better than making an incorrect decision, i.e. $C_{00} < C_{10}$ and $C_{11} < C_{01}$. This implies that $0 < c < 1$.

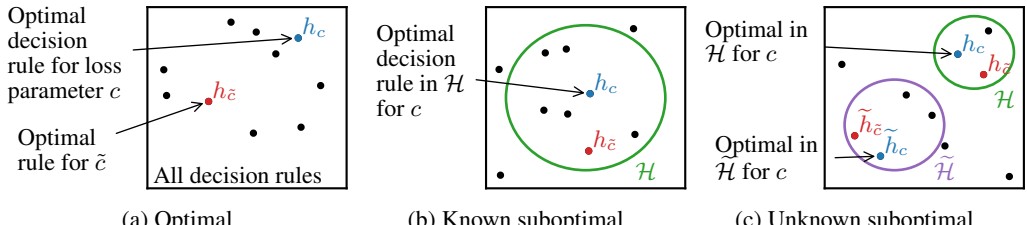

(a) Optimal        (b) Known suboptimal        (c) Unknown suboptimal

Figure 3: We analyze IDT for optimal decision makers and two cases of suboptimal decision makers. (a) In the optimal case (Section 4.1), the decision maker chooses the optimal decision rule $h$ for their loss parameter $c$ from all possible rules. (b) In the known suboptimal case (Section 4.2), the decision maker chooses from a restricted hypothesis class $\mathcal{H}$ which may not contain the overall best decision rule. (c) In the unknown suboptimal case (Section 4.3), the decision maker chooses any of several hypothesis classes $\mathcal{H} \in \mathbb{H}$ and then uses the optimal rule within that class, which may not be the optimal rule amongst all classes. This case is more difficult than (b) because we often need to identify the hypothesis class $\mathcal{H}$ in addition to the loss parameter $c$.

We write $\ell_c$ and $\mathcal{R}_c$ to denote the loss and risk functions using this loss parameter $c$. Thus, we can formally define a binary decision problem:

**Definition 3.2 (Decision problem).** *A (binary) decision problem is a pair $(\mathcal{D}, c)$, where $\mathcal{D}$ is a distribution over pairs of observations and correct decisions $(X, Y) \in \mathcal{X} \times \{0, 1\}$ and $c \in (0, 1)$ is the loss parameter. The decision maker aims to choose a decision rule $h : \mathcal{X} \to \{0, 1\}$ that minimizes the risk $\mathcal{R}_c(h) = \mathbb{E}_{(X,Y) \sim \mathcal{D}}[\ell_c(h(X), Y)]$.*

As a running example, we consider the decision problem where an emergency room (ER) doctor needs to decide whether to treat a patient for a heart attack. In this case, the observation $X$ might consist of the patient's medical records and test results; the correct decision is $Y = 1$ if the patient is having a heart attack and $Y = 0$ otherwise; and the made decision is $\hat{Y} = 1$ if the doctor treats the patient and $\hat{Y} = 0$ if not. In this case, a higher value of $c$ indicates that the doctor places higher cost on accidentally treating a patient not having a heart attack, while a lower value of $c$ indicates the doctor places higher cost on accidentally failing to treat a patient with a heart attack.

In **inverse decision theory (IDT)**, our goal is to determine the loss function the agent is optimizing, which here is equivalent to the parameter $c$. We assume access to the true distribution $\mathcal{D}$ of observations and labels and also a finite sample of observations and decisions $\mathcal{S} = \{(x_1, \hat{y}_1), \ldots, (x_m, \hat{y}_m)\}$ where $x_i \sim \mathcal{D}$ i.i.d. and the decisions are made according to the decision rule, i.e. $\hat{y}_i = h(x_i)$.

Some of our main results concern the effects on IDT of whether or not a decision is made under uncertainty. We now formally characterize such decision problems.

**Definition 3.3 (Decision problems with and without uncertainty).** *A decision problem $(\mathcal{D}, c)$ has no uncertainty if $\mathbb{P}_{(X,Y) \sim \mathcal{D}}(Y = 1 \mid X) \in \{0, 1\}$ almost surely. The decision problem has uncertainty otherwise.*

That is, if it is always the case that, after observing and conditioning on $X$, either $Y = 1$ with 100% probability or $Y = 0$ with 100% probability, then the decision problem has no uncertainty.

# 4 Identifiability and Sample Complexity

We aim to answer two questions about IDT. First, under what assumptions is the loss function identifiable? Second, if the loss function is identifiable, how large must the sample $\mathcal{S}$ be to estimate $c$ to some precision with high probability? We adopt a framework similar to that of probably approximately correct (PAC) learning [27], and aim to calculate a $\hat{c}$ such that with probability at least $1 - \delta$ with respect to the sample of observed decisions, $|\hat{c} - c| \le \epsilon$. While PAC learning typically focuses on test or prediction error, we instead focus on the estimation error for $c$. This has multiple advantages. First, it allows for better understanding and prediction of human behavior across distribution shift or in unseen environments [28]. Second, there are cases where we care about the precise tradeoff the decision maker is optimizing for; for instance, in the ER doctor example, there are guidelines on the tradeoff between different types of treatment errors and we may want

to determine if doctors' behavior aligns with these guidelines [3]. Third, if the decision maker is suboptimal for their loss function (explored in Sections 4.2 and 4.3), we may not want to simply replicate the suboptimal decisions, but find a better decision rule according to the loss function.

We consider three settings where we would like to estimate $c$, illustrated in Figure 3. First, we assume that the decision maker is perfectly optimal for their loss function. This is similar to the framework of Swartz et al. [7]. However, moving beyond their analysis, we present properties necessary for identifiability and sample complexity rates. Second, we relax the assumption that the decision maker is optimal, and instead assume that they only consider a restricted set of hypotheses $\mathcal{H}$ which is known to us. Finally, we remove the assumption that we know the hypothesis class that the decision maker is considering. Instead, we consider a family of hypothesis classes; the decision maker could choose the optimal decision rule *within any class*, which is not necessarily the optimal decision rule *across all classes*.

## 4.1 Optimal decision maker

First, we assume that the decision maker is optimal. In this case, the form of the optimal decision rule is simply the Bayes classifier [29].

**Lemma 4.1 (Bayes optimal decision rule).** *An optimal decision rule $h$ for a decision problem $(\mathcal{D}, c)$ is given by $h(x) = \mathbb{1}\{q(x) \geq c\}$ where $q(x) = \mathbb{P}_{(X,Y)\sim\mathcal{D}}(Y = 1 \mid X = x)$ is the posterior probability of class 1 given the observation $x$.*

That is, any optimal decision rule corresponds to a threshold function on the posterior probability $q(x)$, where the threshold is at the loss parameter $c$. Thus, the strategy for estimating $c$ from a sample of observations and decisions is simple. For each observation $x_i$, we calculate $q(x_i)$. Then, we choose any $\hat{c}$ such that $q(x_i) \geq \hat{c} \Leftrightarrow \hat{y}_i = 1$; that is, $\hat{c}$ is consistent with the observed data. From statistical learning theory, we know that a threshold function can be PAC learned in $O(\log(1/\delta)/\epsilon)$ samples. However, such learning only guarantees low *prediction error* of the learned hypothesis. We need stronger conditions to ensure that $\hat{c}$ is close to the true loss function parameter $c$. The following theorem states conditions which allow estimation of $c$ to arbitrary precision.

**Theorem 4.2 (IDT for optimal decision maker).** *Let $\epsilon > 0$ and $\delta > 0$. Say that there exists $p_c > 0$ such that $\mathbb{P}(q(X) \in (c, c + \epsilon]) \geq p_c\epsilon$ and $\mathbb{P}(q(X) \in [c - \epsilon, c)) \geq p_c\epsilon$. Let $\hat{c}$ be chosen to be consistent with the observed decisions as stated above, i.e. $q(x_i) \geq \hat{c} \Leftrightarrow \hat{y}_i = 1$. Then $|\hat{c} - c| \leq \epsilon$ with probability at least $1 - \delta$ as long as the number of samples $m \geq \frac{\log(2/\delta)}{p_c\epsilon}$.*

The parameter $p_c$ can be interpreted as the approximate probability density of $q(X)$ around the threshold $c$. For instance, the requirements of Theorem 4.2 are satisfied if the random variable $q(X)$ has a probability density of at least $p_c$ on the interval $[c - \rho, c + \rho]$ for some $\rho \geq \epsilon$; the requirements of Theorem 4.2 are more general to allow for cases when $q(X)$ does not have a density. The lower the density $p_c$, and thus the probability of observing decisions close to the threshold $c$, the more difficult inference becomes. Because of this, Theorem 4.2 *requires* that the decision problem has uncertainty. If the decision problem has no uncertainty according to Definition 3.3, then $q(X) \in \{0, 1\}$ always, i.e. the distribution of posterior probabilities has mass only at 0 and 1. In this case, $p_c = 0$ for small enough $\epsilon$ and Theorem 4.2 cannot be applied. In fact, as we show in Section 4.4, it is impossible to tell what the true loss parameter $c$ when the decision problem lacks uncertainty. Figure 2(a-b) illustrates these results.

## 4.2 Suboptimal decision maker with known hypothesis class

Next, we consider cases where the decision maker may not be optimal with respect to their loss function. Our model of suboptimality is that the agent only considers decision rules within some hypothesis class $\mathcal{H}$, which may not include the optimal decision rule. This formulation is similar to that of agnostic PAC learning [10, 11]. It can also be considered a case of a restricted "choice set" as defined in the preference learning literature [30, 31]. It can encompass many types of irrationality or suboptimality. For instance, one could assume that the decision maker is ignoring some of the features in $x$; then $\mathcal{H}$ would consist of only decision rules depending on the remaining features. In the ER doctor example, we might assume that $\mathcal{H}$ consists of decision rules using only the patient's blood pressure and heart rate; this models a suboptimal doctor who is unable to use more data to make a treatment decision.

While there are many possible models of suboptimality, this one has distinct advantages for preference learning with IDT. One alternative model is that the decision maker has small excess risk, i.e. $\mathcal{R}_c(h) \leq \mathcal{R}_c(h^*) + \Delta$ for some small $\Delta$ where $h^*$ is the optimal decision rule. However, this definition precludes identifiability even in the infinite sample limit (see Appendix C). Another form of suboptimality could be that the decision maker chooses a decision rule to minimize a surrogate loss rather than the true loss. However, we show in Appendix F that for reasonable surrogate losses this is no different from minimizing the true loss. A final alternative model of suboptimality is that the human is noisily optimal; this assumption underlies models like Boltzmann rationality or the Shephard-Luce choice rule [32, 26, 33, 14]. However, these models assume stochastic decision making and also cannot handle *systematically* suboptimal humans.

In this section we begin by assuming that the restricted hypothesis class $\mathcal{H}$ is known; this requires some novel analysis but the resulting identifiability conditions and sample complexity are very similar to the optimal case in Section 4.1. In the next section, we consider cases where we are unsure about which restricted hypothesis class the decision maker is considering.

**Definition 4.3.** *A hypothesis class $\mathcal{H}$ is* monotone *if for any $h, h' \in \mathcal{H}$, either $h(x) \geq h'(x) \; \forall x \in \mathcal{X}$ or $h(x) \leq h'(x) \; \forall x \in \mathcal{X}$.*

**Definition 4.4.** *The* optimal subset *of a hypothesis class $\mathcal{H}$ for a distribution $\mathcal{D}$ is defined as*

$$opt_{\mathcal{D}}(\mathcal{H}) = \{h \in \mathcal{H} \mid \exists c \text{ such that } h \in \arg\min_{h \in \mathcal{H}} \mathcal{R}_c(h)\}$$

In this section, we consider hypothesis classes whose optimal subsets are monotone. That is, changing the parameter $c$ has to either flip the optimal decision rule's output for some observations from 0 to 1, or flip some decisions from 1 to 0. It cannot both change some decisions from 0 to 1 and some from 1 to 0. This assumption is mainly technical; many interesting hypothesis classes naturally have monotone optimal subsets. Any hypothesis class formed by thresholding a function is monotone, i.e $\mathcal{H} = \{h(x) = \mathbb{1}\{f(x) \geq b\} \mid b \in \mathbb{R}\}$. Also, the set of decision rules based on a particular subset of the observed features satisfies this criterion, since optimal decision rules in this set are thresholds on the posterior probability that $Y = 1$ given the subset of features.

For hypothesis classes with monotone optimal subsets, we can prove properties that allow for similar analysis to that we introduced in Section 4.1. Let $h_c$ denote a decision rule which is optimal for loss parameter $c$ in hypothesis class $\mathcal{H}$. That is, $h_c \in \arg\min_{h \in \mathcal{H}} \mathcal{R}_c(h)$. A key lemma allows us to define a value similar to the posterior probability we used for analyzing the optimal decision maker.

**Lemma 4.5 (Induced posterior probability).** *Let $opt_{\mathcal{D}}(\mathcal{H})$ be monotone and define*

$$\overline{q}_{\mathcal{H}}(x) \triangleq \sup\left(\{c \in [0,1] \mid h_c(x) = 1\} \cup \{0\}\right) \quad and \quad \underline{q}_{\mathcal{H}}(x) \triangleq \inf\left(\{c \in [0,1] \mid h_c(x) = 0\} \cup \{1\}\right).$$

*Then for all $x \in \mathcal{X}$, $\overline{q}_{\mathcal{H}}(x) = \underline{q}_{\mathcal{H}}(x)$. Define the* induced posterior probability *of $\mathcal{H}$ as $q_{\mathcal{H}}(x) \triangleq \overline{q}_{\mathcal{H}}(x) = \underline{q}_{\mathcal{H}}(x)$.*

**Corollary 4.6.** *Let $h_c$ be any optimal decision rule in $\mathcal{H}$ for loss parameter $c$. Then for any $x \in \mathcal{X}$, $h_c(x) = 1$ if $q_{\mathcal{H}}(x) > c$ and $h_c(x) = 0$ if $q_{\mathcal{H}}(x) < c$.*

Using Lemma 4.5, the problem of IDT again reduces to learning a threshold; this time, any optimal classifier in $\mathcal{H}$ is a threshold function on the *induced* posterior probability $q_{\mathcal{H}}(X)$, as shown in Corollary 4.6. Thus, to estimate $\hat{c}$, we calculate an induced posterior probability $q_{\mathcal{H}}(x_i)$ for each observation $x_i$ and choose any estimate $\hat{c}$ such that $q_{\mathcal{H}}(x_i) \geq \hat{c} \Leftrightarrow \hat{y}_i = 1$. This allows us to state a theorem equivalent to Theorem 4.2 for the suboptimal case.

**Theorem 4.7 (Known suboptimal decision maker).** *Let $\epsilon > 0$ and $\delta > 0$, and let $opt_{\mathcal{D}}(\mathcal{H})$ be monotone. Say that there exists $p_c > 0$ such that $\mathbb{P}(q_{\mathcal{H}}(X) \in (c, c+\epsilon]) \geq p_c \epsilon$ and $\mathbb{P}(q_{\mathcal{H}}(X) \in [c-\epsilon, c)) \geq p_c \epsilon$. Let $\hat{c}$ be chosen to be consistent with the observed decisions, i.e. $q_{\mathcal{H}}(x_i) \geq \hat{c} \Leftrightarrow \hat{y}_i = 1$. Then $|\hat{c} - c| \leq \epsilon$ with probability at least $1 - \delta$ as long as the number of samples $m \geq \frac{\log(2/\delta)}{p_c \epsilon}$.*

## 4.3 Suboptimal decision maker with unknown hypothesis class

We now analyze the case when the decision maker is suboptimal but we are not sure in what manner. We model this by considering a family of hypothesis classes $\mathbb{H}$. We assume that the decision maker considers one of these hypothesis classes $\mathcal{H} \in \mathbb{H}$ and then chooses a rule $h \in \arg\min_{h \in \mathcal{H}} \mathcal{R}_c(h)$. This case is more challenging because we may need to identify $\mathcal{H}$ to identify $c$.

One natural family $\mathbb{H}$ consists of hypothesis classes which depend only on some subset of the features:

$$\mathbb{H}_{\text{feat}} \triangleq \{\mathcal{H}_S \mid S \subseteq \{1, \ldots, n\}\} \quad \text{where} \quad \mathcal{H}_S \triangleq \left\{h(x) = f(x_S) \mid f : \mathbb{R}^{|S|} \to \{0, 1\}\right\} \quad (1)$$

where $x_S$ denotes only the coordinates of $x$ which are in the set $S$. This models a situation where we believe the decision maker may be ignoring some features, but we are not sure which features are being ignored. Another possibility for $\mathbb{H}$ is thresholded linear combinations of the features in $x$, i.e.

$$\mathbb{H}_{\text{linear}} \triangleq \{\mathcal{H}_w \mid w \in \mathbb{R}^n\} \quad \text{where} \quad \mathcal{H}_w \triangleq \left\{h(x) = \mathbb{1}\{w^\top x \geq b\} \mid b \in \mathbb{R}\right\}.$$

In this case, we assume that the decision maker chooses some weights $w$ for the features arbitrarily but then thresholds the combination optimally. This could model the decision maker under- or over-weighting certain features, or also ignoring some (if $w_j = 0$ for some $j$).

In the high pressure and hectic environment of the ER example, we might assume that the doctor is using only a few pieces of data to decide whether to treat a patient. Here, $\mathbb{H}_{\text{feat}}$ would consist of a hypothesis class with decision rules that depend only on blood pressure and heart rate, a hypothesis class with decision rules that rely on these and also on an ECG, and so on. The difficulty of this setting compared to that of Section 4.2 is that the doctor could be using an optimal decision rule within any of these hypothesis classes. Thus, we may need to identify what data the doctor is using in their decision rule in order to identify their loss parameter $c$.

Estimating the loss parameter $c$ in the unknown hypothesis class case requires an additional assumption on the family of hypothesis classes $\mathbb{H}$, in addition to the monotonicity assumption from Section 4.2.

**Definition 4.8.** *Consider a family of hypothesis classes $\mathbb{H}$. Let $h \in \mathcal{H} \in \mathbb{H}$ and $\tilde{\mathcal{H}} \in \mathbb{H}$. Then the minimum disagreement between $h$ and $\tilde{\mathcal{H}}$ is defined as $MD(h, \tilde{\mathcal{H}}) \triangleq \inf_{\tilde{h} \in \tilde{\mathcal{H}}} \mathbb{P}\big(\tilde{h}(X) \neq h(X)\big)$.*

**Definition 4.9.** *A family of hypothesis classes $\mathbb{H}$ and hypothesis $h_c \in \mathcal{H} \in \mathbb{H}$ such that $h_c \in \arg\min_{h \in \mathcal{H}} \mathcal{R}_c(h)$ is $\alpha$-MD-smooth if $opt_{\mathcal{D}}(\tilde{\mathcal{H}})$ is monotone for every $\tilde{\mathcal{H}} \in \mathbb{H}$ and*

$$\forall \tilde{\mathcal{H}} \in \mathbb{H} \quad \forall c' \in (0, 1) \qquad MD(h_{c'}, opt_{\mathcal{D}}(\tilde{\mathcal{H}})) \leq (1 + \alpha|c' - c|)MD(h_c, opt_{\mathcal{D}}(\tilde{\mathcal{H}})).$$

While MD-smoothness is not particularly intuitive at first, it is necessary in some cases to ensure identifiability of the loss parameter $c$. We present a case in Appendix D.2 where a lack of MD-smoothness precludes identifiability.

**Theorem 4.10 (Unknown suboptimal decision maker).** *Let $\epsilon > 0$ and $\delta > 0$. Suppose we observe decisions from a decision rule $h_c$ which is optimal for loss parameter $c$ in hypothesis class $\mathcal{H} \in \mathbb{H}$. Let $h_c$ and $\mathbb{H}$ be $\alpha$-MD-smooth. Furthermore, assume that there exists $p_c > 0$ such that for any $\rho \leq \epsilon$, $\mathbb{P}(q_{\mathcal{H}}(X) \in (c, c + \rho)) \geq p_c \rho$ and $\mathbb{P}(q_{\mathcal{H}}(X) \in (c - \rho, c)) \geq p_c \rho$. Let $d \geq VCdim(\cup_{\mathcal{H} \in \mathbb{H}} \mathcal{H})$ be an upper bound on the VC-dimension of the union of all the hypothesis classes in $\mathbb{H}$.*

*Let $\hat{h}_{\hat{c}} \in \arg\min_{\hat{h} \in \hat{\mathcal{H}}} \mathcal{R}_{\hat{c}}(\hat{h})$ be chosen to be consistent with the observed decisions, i.e. $\hat{h}_{\hat{c}}(x_i) = \hat{y}_i$ for $i = 1, \ldots, m$. Then $|\hat{c} - c| \leq \epsilon$ with probability at least $1 - \delta$ as long as the number of samples $m \geq \tilde{O}\left[\left(\frac{\alpha}{\epsilon} + \frac{1}{\epsilon^2}\right)\left(\frac{d + \log(1/\delta)}{p_c}\right)\right].$*

Theorem 4.10 requires more decision samples to guarantee low estimation error $|\hat{c} - c|$. Unlike Theorems 4.2 and 4.7, the number of samples needed grow with the square of the desired precision $1/\epsilon^2$. There is also a dependence on the VC-dimension of the hypothesis classes $\mathcal{H} \in \mathbb{H}$, since we are not sure which one the decision maker is considering.

Since our results in this section are highly general, it may be difficult to see how they apply to concrete cases. In Appendix E, we explore the specific case of IDT in the unknown hypothesis class setting for $\mathbb{H}_{\text{feat}}$ as defined in (1). We give sufficient conditions for MD-smoothness to hold and show that the sample complexity grows only logaramithically with $n$, the dimension of the observation space $\mathcal{X}$, if the decision maker is relying on a sparse set of features.

## 4.4 Lower bounds

Is there any algorithm which can always determine the loss parameter $c$ to precision $\epsilon$ with high probability using fewer samples than required by Theorems 4.2 and 4.7? We show that the answer

is no: our previously given sample complexity rates are minimax optimal up to constant factors. We formalize this by considering any generic IDT algorithm, which we represent as a function $\hat{c} : (\mathcal{X} \times \{0,1\})^m \rightarrow (0,1)$. The algorithm maps the sample of observations and decisions $\mathcal{S}$ to an estimated loss parameter $\hat{c}(\mathcal{S})$. The algorithm also takes as input the distribution $\mathcal{D}$ and in the suboptimal cases the hypothesis class $\mathcal{H}$ or family of hypothesis classes $\mathbb{H}$, but we leave this dependence implicit in our notation. First, we consider the optimal (Theorem 4.2) and known suboptimal (Theorem 4.7) cases; since these are nearly identical, we focus on the optimal case.

**Theorem 4.11 (Lower bound for optimal decision maker).** *Fix $0 < \epsilon < 1/4$, $0 < \delta \leq 1/2$, and $0 < p_c \leq 1/8\epsilon$. Then for any IDT algorithm $\hat{c}(\cdot)$, there exists a decision problem $(\mathcal{D}, c)$ satisfying the conditions of Theorem 4.7 such that $m < \frac{\log(1/2\delta)}{8p_c\epsilon}$ implies that $\mathbb{P}(|\hat{c}(\mathcal{S}) - c| \geq \epsilon) > \delta$.*

**Corollary 4.12 (Lack of uncertainty precludes identifiability).** *Fix $0 < \epsilon < 1/4$ and suppose a decision problem $(\mathcal{D}, c)$ has no uncertainty. Then for any IDT algorithm $\hat{c}(\cdot)$, there is a loss parameter $c$ and hypothesis class $\mathcal{H}$ such that for any sample size $m$, $\mathbb{P}(|\hat{c}(\mathcal{S}) - c| \geq \epsilon) \geq 1/2$.*

Corollary 4.12 shows that a lack of uncertainty in the decision problem means that no algorithm can learn the loss parameter $c$ to a non-trivial precision with high probability. Thus, uncertainty is *required* for IDT to learn the loss parameter $c$. Since $c$ represents the preferences of the decision maker, *decisions made under certainty do not reveal precise preference information*. In Appendix D, we explore lower bounds for the unknown suboptimal case (Section 4.3 and Theorem 4.10).

## 5    Discussion

Now that we have thoroughly analyzed IDT, we explore its applications, implications, and limitations.

### 5.1    IDT for fine-grained loss functions with applications to fairness

First, we discuss an extension of IDT to loss functions which depend not only on the chosen decision $\hat{Y} = h(X)$ and the ground truth $Y$, but on the observation $X$ as well. In particular, we extend the formulation of IDT from Section 3 to include loss functions which depend on the observations via a "sensitive attribute" $A \in \mathcal{A}$. We denote the value of the sensitive attribute for an observation $x$ by $a(x)$. We again assume that the decision maker chooses the optimal decision rule for this extended loss function:

$$h \in \arg\min_h \mathbb{E}_{(X,Y)\sim\mathcal{D}}[\ell(h(X), Y, a(X))]. \tag{2}$$

This optimal decision rule $h \in \mathcal{H}$ is equivalent to a set of decision rules for every value of $A$, each of which is chosen to minimize the conditional risk for observations with that attribute value:

$$h(x) = h_{a(x)}(x) \quad \text{where} \quad h_a \in \arg\min_h \mathbb{E}_{(X,Y)\sim\mathcal{D}}[\ell(h(X), Y, a) \mid a(X) = a].$$

In this formulation, each attribute-specific decision rule $h_a$ minimizes an expected loss which only depends on the made and correct decisions $h(X)$ and $Y$ over a conditional distribution. Thus, we can split a sample of decisions into samples for each value of the sensitive attribute and perform IDT separately. This will result in a loss parameter estimate $\hat{c}_a$ for each value of $a$.

Once we have estimated loss parameters for each value of $A$, we may ask if the decision maker is applying the same loss function across all such values, i.e. if $c_a = c_{a'}$ for any $a, a' \in \mathcal{A}$. If the loss function is not identical for all values of $A$, i.e. if $c_a \neq c_{a'}$, then one might conclude that the decision maker is unfair or discriminatory against observations with certain values of $A$. For instance, in the ER example, we might be concerned if the doctor is using different loss functions for patients with and without insurance. Concepts like these have received extensive treatment in the machine learning fairness literature, which studies criteria for when a decision rule can be considered "fair." One such fairness criterion is that of group calibration, also known as sufficiency [34, 35, 36]:

**Definition 5.1.** *A decision rule $h : \mathcal{X} \rightarrow \{0,1\}$ for a distribution $(X, Y) \sim \mathcal{D}$ satisfies the* group calibration/sufficiency *fairness criterion if there is a function $r : \mathcal{X} \rightarrow \mathbb{R}$ and threshold $t \in \mathbb{R}$ such that $h(x) = \mathbb{1}\{r(x) \geq t\}$ and $r$ satisfies $Y \perp\!\!\!\perp A \mid r(X)$.*

Testing for group calibration is known to be difficult because of the problem of infra-marginality [37]. While complex Bayesian models have previously been used to perform a "threshold test" for group calibration, we can use IDT to directly test this criterion in an observed decision maker:

**Lemma 5.2 (Equal loss parameters imply group calibration).** *Let $h$ be chosen as in (2) where $\ell(\hat{y}, y, a) = c_a$ if $\hat{y} = 1$ and $y = 0$, $\ell(\hat{y}, y, a) = 1 - c_a$ if $\hat{y} = 0$ and $y = 1$, and $\ell(\hat{y}, y, a) = 0$ otherwise. Then $h$ satisfies group calibration (sufficiency) if $c_a = c_{a'}$ for every $a, a' \in \mathcal{A}$.*

*Conversely, if there exist $a, a' \in \mathcal{A}$ such that $c_a \neq c'_a$ and $\mathbb{P}(q(X) \in (c_a, c_{a'})) > 0$, then $h$ does not satisfy group calibration.*

If we can estimate $c_a$ for a decision rule $h$ for each $a \in \mathcal{A}$, then Lemma 5.2 allows us to immediately determine if $h$ satisfies sufficiency. The minimax guarantees on the accuracy of IDT may make this approach more attractive than the Bayesian threshold test in many scenarios.

### 5.2 Suboptimal decision making with and without uncertainty

We have so far compared the effect of decisions made with and without uncertainty on the *identifiability* of preferences; here, we argue that uncertainty also allows for much more expressive models of *suboptimality* in decision making. In decisions made with certainty, suboptimality can generally only take two forms: either the decision maker is noisy and sometimes randomly makes incorrect decisions, or the decision maker is systematically suboptimal and always makes the wrong decision. Neither seems realistic in the ER doctor example: we would not expect to the doctor to randomly choose not to treat some patients who are clearly having heart attacks, and certainly not expect them to *never* treat patients having heart attacks. In contrast, the models of suboptimality we have presented for uncertain decisions allow for much more rich and realistic forms of suboptimal decision making, like ignoring certain data or over-/under-weighting evidence. We expect that there are similarly more rich forms of suboptimality for uncertain sequential decision problems.

### 5.3 Limitations and future work

While this study sheds significant light on preference learning for uncertain humans, there are some limitations that may be addressed by future work. First, while we assume the data distribution $\mathcal{D}$ of observations $X$ and ground truth decisions $Y$ is known, this is rarely satisfied in practice. However, statistics is replete with methods for estimating properties of a data distribution given samples from it. Such methods are beyond the scope of this work, which focuses on the less-studied problem of inferring a decision maker's loss function. Our work also lacks computational analysis of algorithms for performing IDT. However, such algorithms are likely straightforward; we decide to focus on the statistical properties of IDT, which are more relevant for preference learning in general. Finally, we assume in this work that the decision maker is maximizing expected utility (EU), or equivalently minimizing expected loss. In reality, human decision making may not agree with EU theory; alternative models of decision making under uncertainty such as prospect theory are discussed in the behavioral economics literature [38]. Some work has applied these models to statistical learning [39], but we leave their implications for IDT to future work.

## 6   Conclusion and Societal Impact

We have presented an analysis of preference learning for uncertain humans through the setting of inverse decision theory. Our principle findings are that decisions made under uncertainty can reveal more preference information than obvious ones; and, that uncertainty can alleviate underspecification in preference learning, even in the case of suboptimal decision making. We hope that this and other work on preference learning will lead to AI systems which better understand human preferences and can thus better fulfill them. However, improved understanding of humans could also be applied by malicious actors to manipulate people or invade their privacy. Additionally, building AI systems which learn from human decisions could reproduce racism, sexism, and other harmful biases which are widespread in human decision-making. Despite these concerns, understanding human preferences is important for the long-term positive impact of AI systems. Our work shows that uncertain decisions can be a valuable source of such preference information.

## Acknowledgments and Disclosure of Funding

We would like to thank Kush Bhatia for valuable discussions, Meena Jagadeesan, Sam Toyer, and Alex Turner for feedback on drafts, and the NeurIPS reviewers for helping us improve the clarity of

the paper. This research was supported by the Open Philanthropy Foundation. Cassidy Laidlaw is also supported by a National Defense Science and Engineering Graduate (NDSEG) Fellowship.

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
