# Appendix

## A   Proofs

### A.1   Proof of Lemma 3.1

**Lemma 3.1 (Equivalence of cost matrices).** *Any cost matrix $C = \left(\begin{smallmatrix} C_{00} & C_{01} \\ C_{10} & C_{11} \end{smallmatrix}\right)$ is equivalent to a cost matrix $C' = \left(\begin{smallmatrix} 0 & 1-c \\ c & 0 \end{smallmatrix}\right)$ where $c = \frac{C_{10}-C_{00}}{C_{10}+C_{01}-C_{00}-C_{11}}$ as long as $C_{10} + C_{01} - C_{00} - C_{11} \neq 0$. That is, there are constants $a, b \in \mathbb{R}$ such that $\mathcal{R}_C(h) = a\mathcal{R}_{C'}(h) + b$ for all $h$.*

*Proof.* Let $a = C_{10} + C_{01} - C_{00} - C_{11}$ and $b = \mathbb{P}(Y = 0)C_{00} + \mathbb{P}(Y = 1)C_{11}$. Then

$$
\begin{aligned}
&\mathcal{R}_C(h) \\
&= \mathbb{P}(h(X) = 0 \wedge Y = 0)C_{00} + \mathbb{P}(h(X) = 1 \wedge Y = 0)C_{10} \\
&\quad + \mathbb{P}(h(X) = 0 \wedge Y = 1)C_{01} + \mathbb{P}(h(X) = 1 \wedge Y = 1)C_{11} \\
&= \mathbb{P}(h(X) = 1 \wedge Y = 0)(C_{10} - C_{00}) + \mathbb{P}(Y = 0)C_{00} \\
&\quad + \mathbb{P}(h(X) = 0 \wedge Y = 1)(C_{01} - C_{11}) + \mathbb{P}(Y = 1)C_{11} \\
&= \mathbb{P}(h(X) = 1 \wedge Y = 0)(C_{10} - C_{00}) + \mathbb{P}(h(X) = 0 \wedge Y = 1)(C_{01} - C_{11}) + b \\
&= (C_{10} + C_{01} - C_{00} - C_{11})\left(\mathbb{P}(h(X) = 1 \wedge Y = 0)\frac{C_{10} - C_{00}}{C_{10} + C_{01} - C_{00} - C_{11}}\right. \\
&\qquad \left. + \mathbb{P}(h(X) = 0 \wedge Y = 1)\frac{C_{01} - C_{11}}{C_{10} + C_{01} - C_{00} - C_{11}}\right) + b \\
&= a(\mathbb{P}(h(X) = 1 \wedge Y = 0)c + \mathbb{P}(h(X) = 0 \wedge Y = 1)(1 - c)) + b \\
&= a\mathcal{R}_{C'}(h) + b.
\end{aligned}
$$

∎

### A.2   Proof of Lemma 4.1

**Lemma 4.1 (Bayes optimal decision rule).** *An optimal decision rule $h$ for a decision problem $(\mathcal{D}, c)$ is given by $h(x) = \mathbb{1}\{q(x) \geq c\}$ where $q(x) = \mathbb{P}_{(X,Y)\sim\mathcal{D}}(Y = 1 \mid X = x)$ is the posterior probability of class 1 given the observation $x$.*

This result is well-known [29] but we include a proof here for completeness.

*Proof.* Let $h(x) = \mathbb{1}\{q(x) \geq c\}$ and let $\tilde{h} : \mathcal{X} \to \{0, 1\}$ be any other decision rule. We will show that not only is $h$ an optimal decision rule, but in fact that if $\mathbb{P}(h(X) \neq \tilde{h}(X) \wedge q(X) \neq c) > 0$, then $\mathcal{R}_c(\tilde{h}) > \mathcal{R}_c(h)$; that is, $\tilde{h}$ is strictly suboptimal. Thus, any optimal decision rule $h^*$ must satisfy $h(x) = h^*(x)$ almost surely except where $q(x) = c$.

First, let's define the *conditional risk of $h$ at $x$*, denoted by $\mathcal{R}_c(h \mid X = x)$:

$$\mathcal{R}_c(h \mid X = x) = c\,\mathbb{P}(h(X) = 1 \wedge Y = 0 \mid X = x) + (1 - c)\,\mathbb{P}(h(X) = 0 \wedge Y = 1 \mid X = x).$$

Note that one of the two terms is always zero, depending on whether $h(X)$ is 0 or 1, since $h(X)$ is deterministic given $X$. The risk of $h$ is the expectation of the conditional risk:

$$\mathcal{R}_c(h) = \mathbb{E}_{X\sim\mathcal{D}_x}[\mathcal{R}_c(h \mid X = x)].$$

We can bound the conditional risk for the optimal decision rule $h$:

$$
\begin{aligned}
\mathcal{R}_c(h \mid x) &= \begin{cases} c\,\mathbb{P}(Y = 0 \mid X = x) & q(x) \geq c \\ (1 - c)\mathbb{P}(Y = 1 \mid X = x) & q(x) < c \end{cases} \\
&= \begin{cases} c(1 - q(x)) & q(x) \geq c \\ (1 - c)q(x) & q(x) < c \end{cases} \\
&\leq c(1 - c). \qquad\qquad\qquad\qquad\qquad (3)
\end{aligned}
$$

Now, consider the conditional risk for the other decision rule $\tilde{h}$ at $x$. First, suppose $\tilde{h}(x) = h(x)$; that is, the decision rule agrees with the optimal one. Then clearly $\mathcal{R}_c(\tilde{h} \mid X = x) = \mathcal{R}_c(h \mid X = x) \leq c(1 - c)$. Next, suppose $q(x) \neq c$ and $\tilde{h}(x) \neq h(x)$. Then

$$
\begin{aligned}
\mathcal{R}_c(\tilde{h} \mid X = x) &= \begin{cases} c\mathbb{P}(Y = 0 \mid X = x) & q(x) < c \\ (1-c)\mathbb{P}(Y = 1 \mid X = x) & q(x) > c \end{cases} \\
&= \begin{cases} c(1 - q(x)) & q(x) < c \\ (1-c)q(x) & q(x) > c \end{cases} \\
&> c(1 - c).
\end{aligned}
\tag{4}
$$

Finally, suppose $q(x) = c$; in this case, it is clear that $\mathcal{R}_c(\tilde{h} \mid X = x) = c(1 - c)$ regardless of what $\tilde{h}(x)$ is. Putting this together, we can break down the risk of $\tilde{h}$ by conditioning on whether $\tilde{h}(x) = h(x)$ or $q(x) = c$:

$$
\begin{aligned}
\mathcal{R}_c(\tilde{h}) &= \mathbb{E}[\mathcal{R}_c(\tilde{h} \mid X = x)] \\
&= \mathbb{E}[\mathcal{R}_c(\tilde{h} \mid X = x) \mid \tilde{h}(X) = h(X) \vee q(X) = c] \, \mathbb{P}(\tilde{h}(X) = h(X) \vee q(X) = c) \\
&\quad + \mathbb{E}[\mathcal{R}_c(\tilde{h} \mid X = x) \mid \tilde{h}(X) \neq h(X) \wedge q(X) \neq c] \, \mathbb{P}(\tilde{h}(X) \neq h(X) \wedge q(X) \neq c) \\
&\stackrel{(i)}{>/\geq} \mathbb{E}[\mathcal{R}_c(\tilde{h} \mid X = x) \mid \tilde{h}(X) = h(X) \vee q(X) = c] \, \mathbb{P}(\tilde{h}(X) = h(X) \vee q(X) = c) \\
&\quad + \mathbb{E}[c(1 - c) \mid \tilde{h}(X) \neq h(X) \wedge q(X) \neq c] \, \mathbb{P}(\tilde{h}(X) \neq h(X) \wedge q(X) \neq c) \\
&\stackrel{(ii)}{\geq} \mathbb{E}[\mathcal{R}_c(h \mid X = x) \mid \tilde{h}(X) = h(X) \vee q(X) = c] \, \mathbb{P}(\tilde{h}(X) = h(X) \vee q(X) = c) \\
&\quad + \mathbb{E}[\mathcal{R}_c(h \mid X = x) \mid \tilde{h}(X) \neq h(X) \wedge q(X) \neq c] \, \mathbb{P}(\tilde{h}(X) \neq h(X) \wedge q(X) \neq c) \\
&= \mathbb{E}[\mathcal{R}_c(h \mid X = x)] \\
&= \mathcal{R}_c(h).
\end{aligned}
$$

(i) uses (4) and (ii) uses (3). The above shows that $\mathcal{R}_c(\tilde{h}) \geq \mathcal{R}_c(h)$ for any decision rule $\tilde{h}$, demonstrating that $h$ must have the lowest risk achievable. Note that (i) is strictly greater as long as $\mathbb{P}(\tilde{h}(X) \neq h(X) \wedge q(X) \neq c) > 0$, validating the claim above that any optimal decision rule must agree with $h$ almost surely except when $q(X) = c$.

∎

## A.3   Proof of Theorem 4.2

**Theorem 4.2 (IDT for optimal decision maker).** *Let $\epsilon > 0$ and $\delta > 0$. Say that there exists $p_c > 0$ such that $\mathbb{P}(q(X) \in (c, c + \epsilon]) \geq p_c\epsilon$ and $\mathbb{P}(q(X) \in [c - \epsilon, c)) \geq p_c\epsilon$. Let $\hat{c}$ be chosen to be consistent with the observed decisions as stated above, i.e. $q(x_i) \geq \hat{c} \Leftrightarrow \hat{y}_i = 1$. Then $|\hat{c} - c| \leq \epsilon$ with probability at least $1 - \delta$ as long as the number of samples $m \geq \frac{\log(2/\delta)}{p_c\epsilon}$.*

*Proof.* Let $h$ denote the decision maker's decision rule. From the proof of Lemma 4.1, we know that the optimality of $h$ means that $h(X) = \mathbb{1}\{q(X) \geq c\}$ almost surely as long as $q(X) \neq c$.

Let $E$ denote the event that we observe $x_i$ and $x_j$ in the sample such that $q(x_i) \in (c, c + \epsilon]$ and $q(x_j) \in [c - \epsilon, c)$:

$$
E \quad = \quad \underbrace{\exists x_i \; q(x_i) \in (c, c + \epsilon]}_{E_1} \quad \wedge \quad \underbrace{\exists x_j \; q(x_i) \in [c - \epsilon, c)}_{E_2}.
$$

First, we will lower bound the probability of $E_1$:

$$
\begin{aligned}
\mathbb{P}(E_1) &= 1 - \mathbb{P}(\forall x_i \; q(x_i) \notin (c, c + \epsilon]) \\
&= 1 - (\mathbb{P}(q(X) \notin (c, c + \epsilon]))^m \\
&= 1 - (1 - \mathbb{P}(q(X) \in (c, c + \epsilon]))^m \\
&\geq 1 - (1 - \epsilon p_c)^m
\end{aligned}
$$

$$\geq 1 - e^{-m\epsilon p_c}$$
$$\geq 1 - e^{-\log(2/\delta)}$$
$$= 1 - \delta/2.$$

Second, we will lower bound the probability of $E_2$:

$$\mathbb{P}(E_2) = 1 - \mathbb{P}(\forall j \; q(x_j) \notin [c - \epsilon, c))$$
$$= 1 - (\mathbb{P}(q(X) \notin [c - \epsilon, c)))^m$$
$$= 1 - (1 - \mathbb{P}(q(X) \in [c - \epsilon, c)))^m$$
$$\geq 1 - (1 - \epsilon p_c)^m$$
$$\geq 1 - e^{-m\epsilon p_c}$$
$$\geq 1 - e^{-\log(2/\delta)}$$
$$= 1 - \delta/2.$$

Putting the above together, we can lower bound the probability of $E$:

$$\mathbb{P}(E) = \mathbb{P}(E_1 \wedge E_2)$$
$$= 1 - \mathbb{P}(\neg E_1 \vee \neg E_2)$$
$$\geq 1 - \mathbb{P}(\neg E_1) - \mathbb{P}(\neg E_2)$$
$$\geq 1 - \delta.$$

Finally, we will show that $E$ implies $|\hat{c} - c| \leq \epsilon$. Suppose $E$ occurs. Then $q(x_i) > c$, so $h(x_i) = \hat{y}_i = 1$. This means that $\hat{c} \leq q(x_i) \leq c + \epsilon$. Also, $q(x_j) < c$, so $h(x_j) = \hat{y}_j = 0$. This means that $\hat{c} > q(x_j) \geq c - \epsilon$. Thus

$$c - \epsilon < \hat{c} \leq c + \epsilon$$
$$|\hat{c} - c| \leq \epsilon.$$

So with probability at least $1 - \delta$, $|\hat{c} - c| \leq \epsilon$. ∎

### A.4 Proof of Lemma 4.5

The proof of Lemma 4.5 depends on another lemma, which will also be useful in the unknown hypothesis class setting. This lemma bounds the conditional probability that the correct decision $Y = 1$ for observations $x$ between the decision boundaries of two optimal decision rules.

**Lemma A.1.** *Suppose $opt_{\mathcal{D}}(\mathcal{H})$ is monotone and let $h_c, h_{c'} \in \mathcal{H}$ be optimal decision rules for loss parameters $c$ and $c'$, respectively, where $c < c'$. Then for every $x \in \mathcal{X}$, $h_{c'}(x) \leq h_c(x)$. Furthermore, assuming $\mathbb{P}(h_c(X) \neq h_{c'}(X)) = \mathbb{P}(h_c(X) = 1 \wedge h_{c'}(X) = 0) > 0$,*

$$c \leq \mathbb{P}(Y = 1 \mid h_c(X) = 1 \wedge h_{c'}(X) = 0) \leq c'.$$

*Proof.* We can write the risk of a decision rule $h$ for cost $c$ as

$$\mathcal{R}_c(h) = c\,\mathbb{P}(h(X) = 1 \wedge Y = 0) + (1 - c)\,\mathbb{P}(h(X) = 0 \wedge Y = 1)$$
$$= c\big[\mathbb{P}(Y = 0) - \mathbb{P}(h(X) = 0 \wedge Y = 0)\big] + (1 - c)\,\mathbb{P}(h(X) = 0 \wedge Y = 1)$$
$$= c\Big(\mathbb{P}(Y = 0) - \big[\mathbb{P}(h(X) = 0) - \mathbb{P}(h(X) = 0 \wedge Y = 1)\big]\Big) + (1 - c)\,\mathbb{P}(h(X) = 0 \wedge Y = 1)$$
$$= c\,\mathbb{P}(Y = 0) - c\,\mathbb{P}(h(X) = 0) + c\,\mathbb{P}(h(X) = 0 \wedge Y = 1)$$
$$\quad + \mathbb{P}(h(X) = 0 \wedge Y = 1) - c\,\mathbb{P}(h(X) = 0 \wedge y = 1)$$
$$= c\,\mathbb{P}(Y = 0) - c\,\mathbb{P}(h(X) = 0) + \mathbb{P}(h(X) = 0 \wedge Y = 1). \tag{5}$$

Since $h_c$ is optimal for $c$, we have

$$\mathcal{R}_c(h_{c'}) - \mathcal{R}_c(h_c) \geq 0. \tag{6}$$

Applying (5) to (6) gives

$$\mathbb{P}(h_{c'}(X) = 0 \wedge Y = 1) - \mathbb{P}(h_c(X) = 0 \wedge Y = 1) - c\left[\mathbb{P}(h_{c'}(X) = 0) - \mathbb{P}(h_c(X) = 0)\right] \geq 0. \tag{7}$$

Now, suppose the lemma does not hold; that is, there is some $x \in \mathcal{X}$ such that $h_{c'}(x) > h_c(x)$. Since $\mathrm{opt}_{\mathcal{D}}(\mathcal{H})$ is monotone, this implies

$$\forall x \in \mathcal{X} \quad h_c(x) \leq h_{c'}(x). \tag{$\star$}$$

Assuming $(\star)$ we have the following two identities:

$$\mathbb{P}(h_c(X) = 0) - \mathbb{P}(h_{c'}(X) = 0) = \mathbb{P}(h_c(X) = 0) \wedge h_{c'}(X) = 1)$$

$$\mathbb{P}(h_c(X) = 0 \wedge Y = 1) - \mathbb{P}(h_{c'}(X) = 0 \wedge Y = 1) = \mathbb{P}(h_c(X) = 0) \wedge h_{c'}(X) = 1 \wedge Y = 1).$$

Plugging these in to (5) gives

$$c\,\mathbb{P}(h_c(X) = 0) \wedge h_{c'}(X) = 1) - \mathbb{P}(h_c(X) = 0) \wedge h_{c'}(X) = 1 \wedge Y = 1) \geq 0$$

$$\mathbb{P}(h_c(X) = 0) \wedge h_{c'}(X) = 1 \wedge Y = 1) \leq c\,\mathbb{P}(h_c(X) = 0) \wedge h_{c'}(X) = 1)$$

$$\frac{\mathbb{P}(h_c(X) = 0) \wedge h_{c'}(X) = 1 \wedge Y = 1)}{\mathbb{P}(h_c(X) = 0) \wedge h_{c'}(X) = 1)} \leq c$$

$$\mathbb{P}(Y = 1 \mid h_c(X) = 0 \wedge h_{c'}(X) = 1) \leq c.$$

This is the first claim of the lemma. Now, we can apply the same set of steps to $\mathcal{R}_{c'}(h_c) - \mathcal{R}_{c'}(h_{c'}) \geq 0$ (i.e., using (5) and the above identities) to obtain

$$c' \leq \mathbb{P}(Y = 1 \mid h_c(X) = 0 \wedge h_{c'}(X) = 1).$$

Combining these two equations implies $c' \leq c$, but we assumed that $c < c'$, so this is a contradiction. Thus, $(\star)$ must be false!

Since $\mathrm{opt}_{\mathcal{D}}(\mathcal{H})$ is monotone, the falsity of $(\star)$ implies that actually,

$$\forall x \in \mathcal{X} \quad h_{c'}(x) \leq h_c(x). \tag{8}$$

Now, we can complete the proof by repeating the above steps using (8) instead of $(\star)$ to obtain

$$c \leq \mathbb{P}(Y = 1 \mid h_c(X) = 1 \wedge h_{c'}(X) = 0) \leq c'.$$

$\blacksquare$

**Lemma 4.5 (Induced posterior probability).** *Let $\mathrm{opt}_{\mathcal{D}}(\mathcal{H})$ be monotone and define*

$$\overline{q}_{\mathcal{H}}(x) \triangleq \sup\left(\{c \in [0,1] \mid h_c(x) = 1\} \cup \{0\}\right) \quad \text{and} \quad \underline{q}_{\mathcal{H}}(x) \triangleq \inf\left(\{c \in [0,1] \mid h_c(x) = 0\} \cup \{1\}\right).$$

*Then for all $x \in \mathcal{X}$, $\overline{q}_{\mathcal{H}}(x) = \underline{q}_{\mathcal{H}}(x)$. Define the* induced posterior probability *of $\mathcal{H}$ as $q_{\mathcal{H}}(x) \triangleq \overline{q}_{\mathcal{H}}(x) = \underline{q}_{\mathcal{H}}(x)$.*

*Proof.* Fix $x \in \mathcal{X}$. Using Lemma A.1, we have that

$$c < c' \quad \Rightarrow \quad h_c(x) \geq h_{c'}(x).$$

That is, $h_c(x)$ is monotone non-increasing in $c$. This is enough to show that $q_{\mathcal{H}}(x)$ is well-defined. Consider three cases:

1. $\forall c, h_c(x) = 1$. In this case, $\overline{q}_{\mathcal{H}}(x) = \sup\{c \in [0,1] \mid h_c(x) = 1\} \cup \{0\} = 1$ and $\underline{q}_{\mathcal{H}}(x) = \inf\{c \in [0,1] \mid h_c(x) = 0\} \cup \{1\} = \inf \emptyset \cup \{1\} = 1$ so $q_{\mathcal{H}}(x) = 1$.

2. $\forall c, h_c(x) = 0$. In this case, $\overline{q}_{\mathcal{H}}(x) = \sup\{c \in [0,1] \mid h_c(x) = 1\} \cup \{0\} = \sup \emptyset \cup \{0\} = 0$ and $\underline{q}_{\mathcal{H}}(x) = \inf\{c \in [0,1] \mid h_c(x) = 0\} \cup \{1\} = 0$ so $q_{\mathcal{H}}(x) = 0$.

3. $\exists c_0, c_1$ such that $h_{c_0}(x) = 0$ and $h_{c_1}(x) = 1$. In this case, neither $\{c \in [0,1] \mid h_c(x) = 1\}$ nor $\{c \in [0,1] \mid h_c(x) = 0\}$ is empty so we have

$$\overline{q}_{\mathcal{H}}(x) = \sup\{c \in [0,1] \mid h_c(x) = 1\}$$
$$\underline{q}_{\mathcal{H}}(x) = \inf\{c \in [0,1] \mid h_c(x) = 0\}.$$

Say $q_{\mathcal{H}}(x)$ is not well-defined; that is,

$$\sup\{c \in [0,1] \mid h_c(x) = 1\} \neq \inf\{c \in [0,1] \mid h_c(x) = 0\}.$$

First, suppose $\sup\{c \in [0,1] \mid h_c(x) = 1\} < \inf\{c \in [0,1] \mid h_c(x) = 0\}$. Then there exists some $c$ for which $h_c(x) \notin \{0,1\}$, which is impossible. So $\sup\{c \in [0,1] \mid h_c(x) = 1\} > \inf\{c \in [0,1] \mid h_c(x) = 0\}$. However, this implies that $\exists c_1 \geq c_0$ such that $h_{c_1}(x) = 1$ but $h_{c_0}(x) = 0$. Since $h_c(x)$ is nonincreasing in $c$, this is a contradiction. Thus $q_{\mathcal{H}}(x) = \overline{q}_{\mathcal{H}}(x) = \underline{q}_{\mathcal{H}}(x)$ is well-defined.

■

**Corollary 4.6.** *Let $h_c$ be any optimal decision rule in $\mathcal{H}$ for loss parameter $c$. Then for any $x \in \mathcal{X}$, $h_c(x) = 1$ if $q_{\mathcal{H}}(x) > c$ and $h_c(x) = 0$ if $q_{\mathcal{H}}(x) < c$.*

*Proof.* Let
$$h_c \in \arg\min_{h \in \mathcal{H}} \mathcal{R}_c(h)$$
be an optimal decision rule in $\mathcal{H}$ for loss parameter $c$.

Fix any $x \in \mathcal{X}$. If $q_{\mathcal{H}}(x) = c$, we don't need to prove anything. If $q_{\mathcal{H}}(x) > c$, then suppose $h_c(x) \neq 1$, i.e. $h_c(x) = 0$. Then
$$\underline{q}_{\mathcal{H}}(x) = \inf\{c' \in [0,1] \mid h_{c'}(x) = 0\} \leq c$$
since $h_c(x) = 0$. However, this is a contradiction since we assumed $q_{\mathcal{H}}(x) > c$. Thus $h_c(x) = 1$.

Now, if $q_{\mathcal{H}}(x) < c$, suppose $h_c(x) \neq 0$, i.e. $h_c(x) = 1$. Then
$$\overline{q}_{\mathcal{H}}(x) = \sup\{c' \in [0,1] \mid h_{c'}(x) = 1\} \geq c.$$
This is also a contradiction since we assumed $q_{\mathcal{H}}(x) < c$, so $h_c(x) = 0$. ■

## A.5   Proof of Theorem 4.7

**Theorem 4.7 (Known suboptimal decision maker).** *Let $\epsilon > 0$ and $\delta > 0$, and let $opt_{\mathcal{D}}(\mathcal{H})$ be monotone. Say that there exists $p_c > 0$ such that $\mathbb{P}(q_{\mathcal{H}}(X) \in (c, c + \epsilon]) \geq p_c\epsilon$ and $\mathbb{P}(q_{\mathcal{H}}(X) \in [c - \epsilon, c)) \geq p_c\epsilon$. Let $\hat{c}$ be chosen to be consistent with the observed decisions, i.e. $q_{\mathcal{H}}(x_i) \geq \hat{c} \Leftrightarrow \hat{y}_i = 1$. Then $|\hat{c} - c| \leq \epsilon$ with probability at least $1 - \delta$ as long as the number of samples $m \geq \frac{\log(2/\delta)}{p_c\epsilon}$.*

*Proof.* Let $h \in \mathcal{H}$ denote the decision maker's decision rule. From Corollary 4.6, we know that $h(x) = \mathbb{1}\{q_{\mathcal{H}}(x) \geq c\}$ as long as $q_{\mathcal{H}}(x) \neq c$.

Let $E$ denote the event that we observe $x_i$ and $x_j$ in the sample such that $q_{\mathcal{H}}(x_i) \in (c, c + \epsilon]$ and $q_{\mathcal{H}}(x_j) \in [c - \epsilon, c)$. An analogous computation to the proof of Theorem 4.2 (Section A.3) shows that if $m \geq \frac{\log(2/\delta)}{p_c\epsilon}$, then $\mathbb{P}(E) \geq 1 - \delta$.

If $E$ occurs, then $h(x_i) = 1$ and so $\hat{c} \leq c + \epsilon$. Also, $h(x_j) = 0$ so $\hat{c} \geq c - \epsilon$. Thus, we have
$$\mathbb{P}(|\hat{c} - c| \leq \epsilon) \geq \mathbb{P}(E) \geq 1 - \delta.$$

■

## A.6   Proof of Theorem 4.10

**Theorem 4.10 (Unknown suboptimal decision maker).** *Let $\epsilon > 0$ and $\delta > 0$. Suppose we observe decisions from a decision rule $h_c$ which is optimal for loss parameter $c$ in hypothesis class $\mathcal{H} \in \mathbb{H}$. Let $h_c$ and $\mathbb{H}$ be $\alpha$-MD-smooth. Furthermore, assume that there exists $p_c > 0$ such that for any $\rho \leq \epsilon$, $\mathbb{P}(q_{\mathcal{H}}(X) \in (c, c + \rho)) \geq p_c\rho$ and $\mathbb{P}(q_{\mathcal{H}}(X) \in (c - \rho, c)) \geq p_c\rho$. Let $d \geq VCdim\left(\cup_{\mathcal{H} \in \mathbb{H}} \mathcal{H}\right)$ be an upper bound on the VC-dimension of the union of all the hypothesis classes in $\mathbb{H}$.*

*Let $\hat{h}_{\hat{c}} \in \arg\min_{\hat{h} \in \hat{\mathcal{H}}} \mathcal{R}_{\hat{c}}(\hat{h})$ be chosen to be consistent with the observed decisions, i.e. $\hat{h}_{\hat{c}}(x_i) = \hat{y}_i$ for $i = 1, \ldots, m$. Then $|\hat{c} - c| \leq \epsilon$ with probability at least $1 - \delta$ as long as the number of samples $m \geq \tilde{O}\left[\left(\frac{\alpha}{\epsilon} + \frac{1}{\epsilon^2}\right)\left(\frac{d + \log(1/\delta)}{p_c}\right)\right]$.*

*Proof.* Specifically, we will prove that $\mathbb{P}(|\hat{c} - c| \leq \epsilon) \geq 1 - \delta$ as long as
$$m \geq O\left[\left(\frac{\alpha}{\epsilon} + \frac{1}{\epsilon^2}\right)\left(\frac{d\log(\alpha/(p_c\epsilon)) + \log(1/\delta)}{p_c}\right)\right]. \tag{9}$$

Throughout the proof, let $h_c \in \arg\min_{h \in \mathcal{H}} \mathcal{R}_c(h)$ be the true decision rule and let $\hat{h}_{\hat{c}} \in \arg\min_{\hat{h} \in \hat{\mathcal{H}}} \mathcal{R}_{\hat{c}}(\hat{\mathcal{H}})$ be the estimated decision rule, i.e. one that agrees with the decisions in the sample of observations $\mathcal{S}$.

First, we use a standard result from PAC learning theory to upper bound the disagreement between the estimated decision rule $\hat{h}_{\hat{c}}$ and the true decision rule $h_c$. In particular, since this is a case of *realizable* PAC learning, i.e. the true decision rule $h_c$ is in one of the hypothesis classes $\mathcal{H} \in \mathbb{H}$, we have that

$$\mathbb{P}(h_c(X) \neq \hat{h}_{\hat{c}}(X)) \leq O\left(\frac{1}{\frac{\alpha}{p_c\epsilon} + \frac{1}{p_c\epsilon^2}}\right) = O\left(\min\left(\frac{p_c\epsilon}{\alpha}, p_c\epsilon^2\right)\right)$$

with probability at least $1 - \delta$ over the drawn sample. This bound follows from Vapnik [40] and Blumer et al. [41] since the set of all possible hypotheses $\cup_{\mathcal{H} \in \mathbb{H}} \mathcal{H}$ has VC-dimension at most $d$, and we observe a sample of $m$ observations $x_i$ and decisions $\hat{y}_i = h_c(x_i)$ where $m$ satisfies (9). In particular, denote

$$r = \mathbb{P}(h_c(X) \neq \hat{h}_{\hat{c}}(X)) \leq \min\left(\frac{p_c\epsilon}{6\alpha}, \frac{p_c\epsilon^2}{36}\right). \tag{10}$$

Next, we show that (10) implies that $|\hat{c} - c| \leq \epsilon$; since (10) holds with probability at least $1 - \delta$, this is enough to complete the proof of Theorem 4.10. We will prove that $\hat{c} - c \leq \epsilon$ given (10). The proof that $c - \hat{c} \leq \epsilon$ is analogous. We require a technical lemma on probability theory:

**Lemma A.2.** *Let $A$, $B$, and $C$ be events in a probability space with $\mathbb{P}(A) > 0$ and $\mathbb{P}(B) > 0$. Then*

$$|\mathbb{P}(C \mid A) - \mathbb{P}(C \mid B)| \leq \frac{\mathbb{P}(A \wedge \neg B) + \mathbb{P}(\neg A \wedge B)}{\min(\mathbb{P}(A), \mathbb{P}(B))}.$$

*Proof of Lemma A.2.* To simply the proof of this lemma, we adopt the boolean algebra notation that $AB$ is equivalent to $A \wedge B$ and $\bar{A}$ is equivalent to $\neg A$. Then we have

$$\begin{aligned}
&|\mathbb{P}(C \mid A) - \mathbb{P}(C \mid B)| \\
&= \left|\frac{\mathbb{P}(AC)}{\mathbb{P}(A)} - \frac{\mathbb{P}(BC)}{\mathbb{P}(B)}\right| \\
&= \left|\frac{\mathbb{P}(ABC) + \mathbb{P}(A\bar{B}C)}{\mathbb{P}(AB) + \mathbb{P}(A\bar{B})} - \frac{\mathbb{P}(ABC) + \mathbb{P}(\bar{A}BC)}{\mathbb{P}(AB) + \mathbb{P}(\bar{A}B)}\right| \\
&= \frac{|\mathbb{P}(ABC)\mathbb{P}(\bar{A}B) + \mathbb{P}(A\bar{B}C)\mathbb{P}(B) - \mathbb{P}(ABC)\mathbb{P}(A\bar{B}) - \mathbb{P}(\bar{A}BC)\mathbb{P}(A)|}{\mathbb{P}(A)\mathbb{P}(B)} \\
&\overset{(i)}{\leq} \frac{\max\left(\mathbb{P}(ABC)\mathbb{P}(\bar{A}B) + \mathbb{P}(A\bar{B}C)\mathbb{P}(B), \mathbb{P}(ABC)\mathbb{P}(A\bar{B}) + \mathbb{P}(\bar{A}BC)\mathbb{P}(A)\right)}{\mathbb{P}(A)\mathbb{P}(B)} \\
&= \max\left(\frac{\mathbb{P}(ABC)\mathbb{P}(\bar{A}B) + \mathbb{P}(A\bar{B}C)\mathbb{P}(B)}{\mathbb{P}(A)\mathbb{P}(B)}, \frac{\mathbb{P}(ABC)\mathbb{P}(A\bar{B}) + \mathbb{P}(\bar{A}BC)\mathbb{P}(A)}{\mathbb{P}(A)\mathbb{P}(B)}\right) \\
&\overset{(ii)}{\leq} \max\left(\frac{\mathbb{P}(B)\mathbb{P}(\bar{A}B) + \mathbb{P}(A\bar{B})\mathbb{P}(B)}{\mathbb{P}(A)\mathbb{P}(B)}, \frac{\mathbb{P}(A)\mathbb{P}(A\bar{B}) + \mathbb{P}(\bar{A}B)\mathbb{P}(A)}{\mathbb{P}(A)\mathbb{P}(B)}\right) \\
&= \max\left(\frac{\mathbb{P}(\bar{A}B) + \mathbb{P}(A\bar{B})}{\mathbb{P}(A)}, \frac{\mathbb{P}(A\bar{B}) + \mathbb{P}(\bar{A}B)}{\mathbb{P}(B)}\right) \\
&= \frac{\mathbb{P}(A\bar{B}) + \mathbb{P}(\bar{A}B)}{\min(\mathbb{P}(A), \mathbb{P}(B))}.
\end{aligned}$$

(i) uses the fact that for positive $u$ and $v$, $|u - v| \leq \max(u, v)$. (ii) uses the fact that $\mathbb{P}(E_1 E_2) \leq \mathbb{P}(E_1)$ for any events $E_1$ and $E_2$. $\qquad\square$

Essentially, Lemma A.2 says that if events $A$ and $B$ have high "overlap," then the conditional probabilities of another event $C$ given $A$ and $B$ should be close. We next carefully construct two such events with high overlap.

First, let $c' = c + \epsilon/2$ and let $h_{c'} \in \arg\min_{h \in \mathcal{H}} \mathcal{R}_{c'}(h)$. Since $h_c$ and $\mathbb{H}$ are $\alpha$-MD-smooth, we have that

$$\mathrm{MD}(h_{c'}, \mathrm{opt}_{\mathcal{D}}(\hat{\mathcal{H}})) \leq (1 + \alpha|c' - c|)\mathrm{MD}(h_c, \mathrm{opt}_{\mathcal{D}}(\hat{\mathcal{H}})) \tag{11}$$

$$\leq (1 + \alpha\epsilon/2)\mathbb{P}(h_c(X) \neq \hat{h}_{\hat{c}}(X))$$
$$\leq (1 + \alpha\epsilon/2)r. \tag{12}$$

Since $\mathrm{MD}(h, \mathrm{opt}_{\mathcal{D}}(\hat{\mathcal{H}})) = \inf_{\hat{h} \in \mathrm{opt}_{\mathcal{D}}(\hat{\mathcal{H}})} \mathbb{P}(h(X) \neq \hat{h}(X))$, there must be some hypothesis $\hat{h}_{\hat{c}'} \in \arg\min_{\hat{h} \in \hat{\mathcal{H}}} \mathcal{R}_{\hat{c}'}(\hat{h})$ that matches the minimum disagreement with $h_{c'}$ plus a small positive number (in case the infimum is not achieved):

$$\mathbb{P}(\hat{h}_{\hat{c}'}(X) \neq h_{c'}(X)) \leq \mathrm{MD}(h_{c'}, \mathrm{opt}_{\mathcal{D}}(\hat{c})) + r$$
$$\leq (2 + \alpha\epsilon/2)r.$$

Now, let the events $A$, $B$, and $C$ be defined as follows:

$$A: \quad h_c(X) = 1 \wedge h_{c'}(X) = 0,$$
$$B: \quad \hat{h}_{\hat{c}}(X) = 1 \wedge \hat{h}_{\hat{c}'}(X) = 0,$$
$$C: \quad Y = 1.$$

Using Lemma A.2, we can write the bound

$$\mathbb{P}(Y = 1 \mid B) \leq \mathbb{P}(Y = 1 \mid A) + \frac{\mathbb{P}(A \wedge \neg B \vee \neg A \wedge B)}{\min(\mathbb{P}(A), \mathbb{P}(B))}. \tag{13}$$

We will establish bounds on each term in (13).

**Upper bound on $\mathbb{P}(A \wedge \neg B \vee \neg A \wedge B)$**    It is easy to see that

$$A \wedge \neg B \vee \neg A \wedge B \quad \Rightarrow \quad h_c(X) \neq \hat{h}_{\hat{c}}(X) \vee h_{c'}(X) \neq \hat{h}_{\hat{c}'}(X).$$

Given this implication, it must be that

$$\mathbb{P}(A \wedge \neg B \vee \neg A \wedge B) \leq \mathbb{P}(h_c(X) \neq \hat{h}_{\hat{c}}(X) \vee h_{c'}(X) \neq \hat{h}_{\hat{c}'}(X))$$
$$\leq (3 + \alpha\epsilon/2)r$$
$$\leq p_c\epsilon^2/12 + p_c\epsilon^2/12 = p_c\epsilon^2/6$$

where the inequalities follow from (10) and (12).

**Lower bound on $\min(\mathbb{P}(A), \mathbb{P}(B))$**    Since $h_c$ is optimal within $\mathcal{H}$ for loss parameter $c$, Corollary 4.6 gives that $h_c(x) = 1$ if $q_{\mathcal{H}}(x) > c$. Similarly, $h_{\hat{c}}(x) = 0$ if $q_{\mathcal{H}}(x) < c$. Therefore,

$$q_{\mathcal{H}}(X) \in (c, c') \quad \Rightarrow \quad h_c(X) = 1 \wedge h_{\hat{c}}(X) = 0 \quad \Leftrightarrow \quad A.$$

This implication allows us to lower bound $\mathbb{P}(A)$:

$$\mathbb{P}(A) \geq \mathbb{P}(q_{\mathcal{H}}(X) \in (c, c')) = \mathbb{P}(q_{\mathcal{H}}(X) \in (c, c + \epsilon/2)) \geq p_c\epsilon/2$$

where the final inequality is by assumption. We also need to lower bound $\mathbb{P}(B)$ in order to lower bound $\min(\mathbb{P}(A), \mathbb{P}(B))$:

$$\mathbb{P}(B) = \mathbb{P}(A \wedge B) + \mathbb{P}(\neg A \wedge B)$$
$$= \mathbb{P}(A) - \mathbb{P}(A \wedge \neg B) + \mathbb{P}(\neg A \wedge B)$$
$$\geq \mathbb{P}(A) - \Big(\mathbb{P}(A \wedge \neg B) + \mathbb{P}(\neg A \wedge B)\Big)$$
$$\geq p_c\epsilon/2 - p_c\epsilon^2/6$$
$$\geq p_c\epsilon/3.$$

We assume that $\epsilon \leq 1$ to lower bound $\epsilon \geq \epsilon^2$, but this is fine since if $\epsilon > 1$ then Theorem 4.10 holds trivially. Thus we have $\min(\mathbb{P}(A), \mathbb{P}(B)) \geq p_c\epsilon/3$.

**Lower bound on $\mathbb{P}(Y = 1 \mid B)$**    By Lemma A.1, we have that, since $\mathbb{P}(B) > 0$,

$$\mathbb{P}(Y = 1 \mid B) = \mathbb{P}(Y = 1 \mid \hat{h}_{\hat{c}}(X) = 1 \wedge \hat{h}_{\hat{c}'}(X) = 0) \geq \hat{c}.$$

**Upper bound on $\mathbb{P}(Y = 1 \mid A)$**    Similarly, by Lemma A.1, we have that, since $c' > c$ and $\mathbb{P}(A) > 0$,

$$\mathbb{P}(Y = 1 \mid A) = \mathbb{P}(Y = 1 \mid h_c(X) = 1 \wedge h_{c'}(X) = 0) \leq c' = c + \epsilon/2.$$

**Concluding the proof**    Given all these bounds, we can rewrite (12) as

$$\hat{c} \leq \mathbb{P}(Y = 1 \mid B) \leq \mathbb{P}(Y = 1 \mid A) + \frac{\mathbb{P}(A \wedge \neg B \vee \neg A \wedge B)}{\min(\mathbb{P}(A), \mathbb{P}(B))}$$

$$\leq c + \epsilon/2 + \frac{p_c \epsilon^2/6}{p_c \epsilon/3}$$

$$\leq c + \epsilon/2 + \epsilon/2 = c + \epsilon$$

$$\hat{c} - c \leq \epsilon.$$

This completes the proof that $\hat{c} - c \leq \epsilon$ with probability at least $1 - \delta$; the proof that $c - \hat{c} \leq \epsilon$ is analogous. $\blacksquare$

## A.7   Proof of Theorem 4.11

**Theorem 4.11 (Lower bound for optimal decision maker).** *Fix $0 < \epsilon < 1/4$, $0 < \delta \leq 1/2$, and $0 < p_c \leq 1/8\epsilon$. Then for any IDT algorithm $\hat{c}(\cdot)$, there exists a decision problem $(\mathcal{D}, c)$ satisfying the conditions of Theorem 4.7 such that $m < \frac{\log(1/2\delta)}{8p_c\epsilon}$ implies that $\mathbb{P}(|\hat{c}(\mathcal{S}) - c| \geq \epsilon) > \delta$.*

*Proof.* Consider a distribution over $X \in \mathcal{X} = [0, 1]$ where

$$q(x) = \mathbb{P}(Y = 1 \mid X = x) = x.$$

Let the distribution $\mathcal{D}_X$ over $X$ have density $p_c$ on the interval $(1/2 - 2\epsilon, 1/2 + 2\epsilon)$ and let $\mathbb{P}(X = 0) = \mathbb{P}(X = 1) = 1/2 - 2p_c\epsilon$.

Let $c_1 = 1/2 - \epsilon$ and $c_2 = 1/2 + \epsilon$. Then clearly, for $c \in \{c_1, c_2\}$, the conditions of Theorem 4.2 are satisfied:

$$\mathbb{P}(q(X) \in [c - \epsilon, c)) = \mathbb{P}(q(X) \in (c, c + \epsilon]) = p_c\epsilon.$$

By Lemma 4.1, the optimal decision rule for loss parameter $c_1$ is $h_{c_1}(x) = \mathbb{1}\{x \geq c_1\}$ and for $c_2$ it is $h_{c_2}(x) = \mathbb{1}\{x \geq c_2\}$.

Now suppose

$$m < \frac{\log(1/2\delta)}{8p_c\epsilon}$$

as stated in the theorem. We can bound the probability of the following event $E$:

$$\mathbb{P}(\underbrace{\forall x_i \in \mathcal{S} \quad q(x_i) \in \{0, 1\}}_{E}) = [\mathbb{P}(X \in \{0, 1\})]^m$$

$$= (1 - 4p_c\epsilon)^m$$

$$\overset{(i)}{\geq} \left(e^{-8p_c\epsilon}\right)^m$$

$$= e^{-\log(1/2\delta)} = 2\delta.$$

(i) uses the fact that $1 - u \geq e^{-2u}$ for $u \in [0, 1/2]$. Now, suppose $E$ occurs. In this case, $h_{c_1}(x_i) = h_{c_2}(x_i)$ for all $x_i \in \mathcal{S}$. That is, regardless of which loss parameter $c \in \{c_1, c_2\}$ is used, the distribution of samples will be the same. Let $\mathcal{S}_1$ denote the random variable for a sample taken from a decision maker using $h_{c_1}$ and $\mathcal{S}_2$ a sample taken from $h_{c_2}$. Since these have the same distribution under $E$, they must induce the same probabilities when the IDT algorithm $\hat{c}$ is applied to them:

$$p_1 = \mathbb{P}(\hat{c}(\mathcal{S}_1) \leq 1/2 \mid E) = \mathbb{P}(\hat{c}(\mathcal{S}_2) \leq 1/2 \mid E),$$
$$p_2 = \mathbb{P}(\hat{c}(\mathcal{S}_1) > 1/2 \mid E) = \mathbb{P}(\hat{c}(\mathcal{S}_2) > 1/2 \mid E).$$

Since $p_1 + p_2 = 1$, at least one of $p_1, p_2 \geq 1/2$. Suppose WLOG that $p_1 \geq 1/2$. Then

$$\mathbb{P}(|\hat{c}(\mathcal{S}_2) - c_2| \geq \epsilon) \geq \mathbb{P}(\hat{c}(\mathcal{S}_2) \leq 1/2)$$
$$= \mathbb{P}(\hat{c}(\mathcal{S}_2) \leq 1/2 \mid E)\,\mathbb{P}(E)$$
$$\geq 1/2(2\delta) = \delta.$$

Thus there is a decision problem $(\mathcal{D}, c_2)$ for which the IDT algorithm $\hat{c}$ must make an error of at least size $\epsilon$ with at least probability $\delta$. This concludes the proof. $\blacksquare$

**Corollary 4.12 (Lack of uncertainty precludes identifiability).** *Fix $0 < \epsilon < 1/4$ and suppose a decision problem $(\mathcal{D}, c)$ has no uncertainty. Then for any IDT algorithm $\hat{c}(\cdot)$, there is a loss parameter $c$ and hypothesis class $\mathcal{H}$ such that for any sample size $m$, $\mathbb{P}(|\hat{c}(\mathcal{S}) - c| \geq \epsilon) \geq 1/2$.*

*Proof.* Let the loss parameters $c_1 = 1/2 - \epsilon$ and $c_2 = 1/2 + \epsilon$ be defined as in the proof of Theorem 4.11 above. By Lemma 4.1, the optimal decision rule for loss parameter $c_1$ is $h_{c_1}(x) = \mathbb{1}\{q(x) \geq c_1\}$ and for $c_2$ it is $h_{c_2}(x) = \mathbb{1}\{q(x) \geq c_2\}$. Since $\mathbb{P}(q(x) \in \{0, 1\}) = 1$, it is clear that the decision rules make the same decision rules almost surely, i.e. $\mathbb{P}(h_{c_1}(X) = h_{c_2}(X)) = 1$. Thus, letting $\mathcal{S}_1$ and $\mathcal{S}_2$ denote samples drawn from decision rules $h_{c_1}$ and $h_{c_2}$, respectively, as above, we have that the distributions of $\mathcal{S}_1$ and $\mathcal{S}_2$ are indistinguishable. Thus by the same argument as above we can show that (WLOG)

$$\mathbb{P}(|\hat{c}(\mathcal{S}_2) - c_2| \geq \epsilon) \geq \mathbb{P}(\hat{c}(\mathcal{S}_2) \leq 1/2) \geq 1/2.$$

∎

## A.8 Proof of Lemma 5.2

**Definition 5.1.** *A decision rule $h : \mathcal{X} \to \{0, 1\}$ for a distribution $(X, Y) \sim \mathcal{D}$ satisfies the* group calibration/sufficiency *fairness criterion if there is a function $r : \mathcal{X} \to \mathbb{R}$ and threshold $t \in \mathbb{R}$ such that $h(x) = \mathbb{1}\{r(x) \geq t\}$ and $r$ satisfies $Y \perp\!\!\!\perp A \mid r(X)$.*

**Lemma 5.2 (Equal loss parameters imply group calibration).** *Let $h$ be chosen as in (2) where $\ell(\hat{y}, y, a) = c_a$ if $\hat{y} = 1$ and $y = 0$, $\ell(\hat{y}, y, a) = 1 - c_a$ if $\hat{y} = 0$ and $y = 1$, and $\ell(\hat{y}, y, a) = 0$ otherwise. Then $h$ satisfies group calibration (sufficiency) if $c_a = c_{a'}$ for every $a, a' \in \mathcal{A}$.*

*Conversely, if there exist $a, a' \in \mathcal{A}$ such that $c_a \neq c'_a$ and $\mathbb{P}(q(X) \in (c_a, c_{a'})) > 0$, then $h$ does not satisfy group calibration.*

*Proof that equal $c_a$ imply group calibration.* Assume $c_a = c_{a'} = c$ for every $a, a' \in \mathcal{A}$. Then define

$$r(x) = q(x) + h(x). \tag{14}$$

That is, $r(x)$ is the posterior probability $q(x) = \mathbb{P}(Y = 1 \mid X = x)$ plus one if the decision rule outputs the decision $h(x) = 1$. From the proof of Lemma 4.1, we know that $h(x) = 1$ if $q(x) > c$ and $h(x) = 0$ if $q(x) < c$. From this and (14) we can write

$$h(x) = \mathbb{1}\{r(x) \geq c + 1\}.$$

Now we need to show that $Y \perp\!\!\!\perp A \mid r(X)$. Note that $r(X) \in [0, c] \cup [c + 1, 2]$. First, we consider $r(X) \in [0, c]$. In this case, for any $a \in \mathcal{A}$, we have

$$\begin{aligned}
\mathbb{P}(Y = 1 \mid A = a, r(X) = r) &= \mathbb{P}(Y = 1 \mid A = a, q(X) = r) \\
&= r \\
&= \mathbb{P}(Y = 1 \mid r(X) = r).
\end{aligned}$$

Next, say $r(X) \in [c + 1, 2]$. Then

$$\begin{aligned}
\mathbb{P}(Y = 1 \mid A = a, r(X) = r) &= \mathbb{P}(Y = 1 \mid A = a, q(X) = r - 1) \\
&= r - 1 \\
&= \mathbb{P}(Y = 1 \mid r(X) = r).
\end{aligned}$$

So in either case, $\mathbb{P}(Y = 1 \mid A = a, r(X) = r) = \mathbb{P}(Y = 1 \mid r(X) = r)$. Thus $Y \perp\!\!\!\perp A \mid r(X)$. ∎

*Proof of inverse.* Now, assume $\exists\, a, a' \in \mathcal{A}$ such that $c_a \neq c_{a'}$. WLOG, suppose that $c_a < c_{a'}$. Let $r : \mathcal{X} \to \mathbb{R}$ be any function satisfying $h(x) = \mathbb{1}\{r(x) \geq t\}$. WLOG we can also assume $t = 0$. From Lemma 4.1, we know that if $a(x) = a$, then $q(x) < c_a$ implies $h(x) = 0$ and $q(x) > c_a$ implies $h(x) = 1$. Also, if $a(x) = a'$, then $q(x) < c_{a'}$ implies $h(x) = 0$ and $q(x) > c_{a'}$ implies $h(x) = 1$. Therefore,

$$\begin{aligned}
&\mathbb{P}(Y = 1 \mid A = a, r(X) > 0) \\
&= \mathbb{P}(Y = 1 \mid A = a, q(X) > c_a) \\
&= \mathbb{P}(Y = 1 \mid q(X) > c_a)
\end{aligned}$$

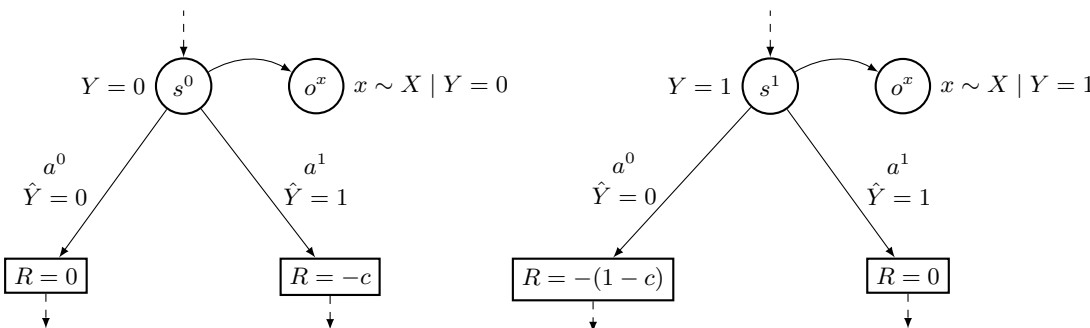

Figure 4: A graphical depiction of the POMDP formulation of IDT described in Appendix B. A state in $\{s^0, s^1\}$ is randomly selected at each timestep and an observation is generated according to the conditional distribution of $X \mid Y$. An action (decision) is taken and the agent receives reward equal to the negative of the loss.

$$
\begin{aligned}
&= \mathbb{P}(Y = 1 \mid q(X) \in (c_a, c_{a'}))\frac{\mathbb{P}(q(X) \in (c_a, c_{a'}))}{\mathbb{P}(q(X) > c_a)} \\
&\quad + \mathbb{P}(Y = 1 \mid q(X) \geq c_{a'})\frac{\mathbb{P}(q(X) \geq c_{a'})}{\mathbb{P}(q(X) > c_a)} \\
&\overset{(i)}{<} c_{a'}\frac{\mathbb{P}(q(X) \in (c_a, c_{a'}))}{\mathbb{P}(q(X) > c_a)} + \mathbb{P}(Y = 1 \mid q(X) \geq c_{a'})\frac{\mathbb{P}(q(X) \geq c_{a'})}{\mathbb{P}(q(X) > c_a)} \\
&\overset{(ii)}{\leq} \mathbb{P}(Y = 1 \mid q(X) \geq c_{a'}) \\
&\leq \mathbb{P}(Y = 1 \mid q(X) > c_{a'}) \\
&= \mathbb{P}(Y = 1 \mid A = a', q(X) > c_{a'}) \\
&= \mathbb{P}(Y = 1 \mid A = a', r(X) > 0).
\end{aligned}
$$

(i) and (ii) make use of the fact that

$$
\begin{aligned}
\mathbb{P}(Y = 1 \mid q(X) \in (c_a, c_{a'})) &= \mathbb{E}[q(X) \mid q(X) \in (c_a, c_{a'})] \\
&< c_{a'} \leq \mathbb{E}[q(X) \mid q(X) \geq c_{a'}] = \mathbb{P}(Y = 1 \mid q(X) \geq c_{a'}).
\end{aligned}
$$

(i) also uses the assumption that $\mathbb{P}(q(X) \in (c_a, c_{a'})) > 0$.

Thus, we have that $\mathbb{P}(Y = 1 \mid A = a, r(X) > 0) \neq \mathbb{P}(Y = 1 \mid A = a', r(X) > 0)$; therefore, $Y$ and $A$ are *not* independent given $r(X)$, so group calibration is not satisfied. ∎

## B  POMDP Formulation of IDT

As mentioned in the main text, IDT can be seen as a special case of inverse reinforcement learning (IRL) in a partially observable Markov decision process (POMDP) (or equivalently, belief state MDP). Here, we present the equivalent POMDP and discuss connections to to our results.

A POMDP is a tuple consisting of seven elements. For an IDT decision problem $(\mathcal{D}, c)$ they are:

- The state space consists of two states, each corresponding to a value of $Y$, the ground truth/correct decision. We call them $s^0$ for $Y = 0$ and $s^1$ for $Y = 1$.

- The action space consists of two actions, each corresponding to one of the decisions $\hat{Y}$. We equivalently call them $a^0$ for $\hat{Y} = 0$ and $a^1$ for $\hat{Y} = 1$.

- The transition probabilities do not depend on the previous state or action; rather, $s^0$ or $s^1$ is randomly selected based on their probabilities under the distribution $\mathcal{D}$:

$$
\begin{aligned}
p(s_{t+1} = s^0 \mid s_t, a_t) &= \mathbb{P}_{X,Y \sim \mathcal{D}}(Y = 0), \\
p(s_{t+1} = s^1 \mid s_t, a_t) &= \mathbb{P}_{X,Y \sim \mathcal{D}}(Y = 1).
\end{aligned}
$$

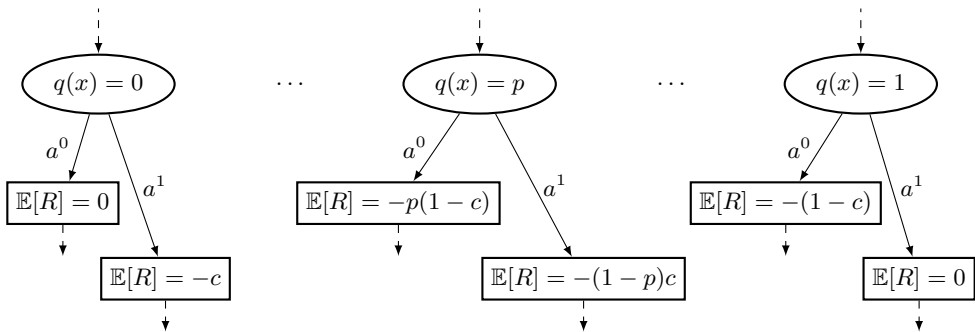

Figure 5: A graphical depiction of the belief state MDP formulation of IDT. There is a belief state for each posterior probability $q(x) = \mathbb{P}(Y = 1 \mid X = x) \in [0, 1]$. Observing the agent at a belief state gives a constraint on their reward function [2]. Thus, if $q(X)$ has support on $[0, 1]$, i.e. if there is a significant range of uncertainty in the decision problem, then there can be arbitrarily many such constraints, allowing the loss parameter $c$ to be learned to arbitrary precision.

- The reward function is the negative of the loss function described in Section 3:

$$R(s^0, a^0) = 0 \qquad\qquad R(s^1, a^0) = -(1 - c),$$
$$R(s^0, a^1) = -c \qquad\qquad R(s^1, a^1) = 0.$$

- The observation space includes elements for each $X \in \mathcal{X}$. We denote by $o^x$ the POMDP observation for $x \in \mathcal{X}$.
- The observation probabilities are

$$p(o_t = o^x \mid s_t = s^y) = \mathbb{P}(X = x \mid Y = y).$$

- The discount factor $\gamma$ is basically irrelevant to IDT, since the decisions are non-sequential. Thus any $\gamma$ will produce the same behavior.

A graphical depiction of this POMDP is shown in Figure 4. Any decision rule $h : \mathcal{X} \to \{0, 1\}$ corresponds to a policy $\pi$ in this POMDP:

$$\pi(a_t = a^{\hat{y}} \mid o_t = o^x) = \mathbb{1}\{h(x) = \hat{y}\}.$$

**Belief state MDP**   The above POMDP can be equivalently formulated as a belief state MDP. The belief states correspond to values of the posterior probability

$$\mathbb{P}(s = s^1 \mid o = o^x) = \mathbb{P}(Y = 1 \mid X = x) = q(x).$$

A graphical depiction of this belief state reduction is shown in Figure 5.

Since the POMDP is non-sequential, these beliefs only depend on the most recent observation $o^x$. The expected reward for action $a^{\hat{y}}$ at belief state with posterior probability $q(x)$ is

$$R(q(x), a^0) = \mathbb{P}(s = s^0 \mid q(x))R(s^0, a^0) + \mathbb{P}(s = s^1 \mid q(x))R(s^1, a^0) = -q(x)(1 - c),$$
$$R(q(x), a^1) = \mathbb{P}(s = s^0 \mid q(x))R(s^0, a^1) + \mathbb{P}(s = s^1 \mid q(x))R(s^1, a^1) = -(1 - q(x))c.$$

Thus, observing decision $a^0$ at a belief state $q(x)$ indicates that

$$R(q(x), a^0) \geq R(q(x), a^1)$$
$$-q(x)(1 - c) \geq -(1 - q(x))c$$
$$c \geq q(x).$$

Similarly, observing decision $a^1$ at a belief state $q(x)$ indicates that

$$R(q(x), a^0) \leq R(q(x), a^1)$$

$$-q(x)(1 - c) \leq -(1 - q(x))c$$
$$c \leq q(x).$$

Thus, as described in Section 4.1, IDT in this (optimal) case consists of determining the threshold on $q(x)$ where the action switches from $a^0$ to $a^1$ for observations $o^x$.

This formulation gives some additional insight into why uncertainty is helpful for IDT. If $q(x) \in \{0, 1\}$ always, then there are only two belief states corresponding to $q(x) = 0$ and $q(x) = 1$. Thus, we only obtain two constraints on the value of $c$, i.e. $0 \leq c \leq 1$. However, if $q(X)$ has support on all of $[0, 1]$, then we there belief states corresponding to every $q(x) \in [0, 1]$. Thus we can obtain infinite constraints on the value of $c$, allowing learning it to arbitrary precision as shown in Section 4.1.

## C  Alternative Suboptimality Model

As mentioned in Section 4.2, there are many ways to model suboptimal decision making. One possibility is to only require that the decision rule $h$ is *close* to optimal, i.e.

$$\mathcal{R}_c(h) \leq \mathcal{R}_c^{\text{opt}} + \Delta \qquad \text{where} \qquad \mathcal{R}_c^{\text{opt}} = \inf_{h^*} \mathcal{R}_c(h^*). \tag{15}$$

However, as we show in the following lemma, this assumption can preclude identifiablity of $c$. The models of suboptimality we present in Sections 4.2 and 4.3, in contrast, still allow exact identifiability of the loss parameter.

**Lemma C.1 (Loss cannot always be identified for close-to-optimal decision rules).** *Fix $0 < \Delta \leq 1$ and $0 < \epsilon < 1/4$. Then for any IDT algorithm $\hat{c}(\cdot)$, there is a decision problem $(\mathcal{D}, c)$ and a decision rule $h$ which is $\Delta$-close to optimal as in (15) such that*

$$\mathbb{P}(|\hat{c}(\mathcal{S}) - c| \geq \epsilon) \geq 1/2,$$

*where the sample $\mathcal{S}$ of any size $m$ is observed from the decision rule $h$. Furthermore, the distribution $\mathcal{D}$ and loss parameter $c$ satisfy the requirements of Theorem 4.2 for when the decision maker is optimal.*

*Proof.* Consider a distribution over $X \in \mathcal{X} = [0, 1]$ where

$$q(x) = \mathbb{P}(Y = 1 \mid X = x) = x.$$

Let the distribution $\mathcal{D}_X$ have density $\Delta$ on the interval $(1/2 - 2\epsilon, 1/2 + 2\epsilon)$ and let $\mathbb{P}(X = 0) = \mathbb{P}(X = 1) = 1/2 - 2\Delta\epsilon$.

Let $c_1 = 1/2 - \epsilon$ and $c_2 = 1/2 + \epsilon$. Then clearly $\mathbb{P}(q(X) \in [c - \epsilon, c)) = \mathbb{P}(q(X) \in (c, c + \epsilon]) = \epsilon\Delta$ for $c \in \{c_1, c_2\}$. Thus either $c_1$ or $c_2$ satisfies the conditions of Theorem 4.2.

Now define identical decision rules

$$h_1(x) = h_2(x) = \mathbb{1}\{x \geq 1/2 - \epsilon\}.$$

From Lemma 4.1, we know that $h_1$ is optimal for $c_1$, so it is certainly $\Delta$-close to optimal. We can show that $h_2$ is $\Delta$-close to optimal for $c_2$ as well:

$$\mathcal{R}_{c_2}(h_2) - \mathcal{R}_{c_2}(x \mapsto \mathbb{1}\{x \geq 1/2 + \epsilon\})$$
$$= \mathbb{E}\Big[\ell(\mathbb{1}\{X \geq 1/2 - \epsilon\}, Y) - \ell(\mathbb{1}\{X \geq 1/2 + \epsilon\}, Y)\Big]$$
$$= \mathbb{E}\Big[\ell(\mathbb{1}\{X \geq 1/2 - \epsilon\}, Y) - \ell(\mathbb{1}\{X \geq 1/2 + \epsilon\}, Y) \mid X \in [1/2 - \epsilon, 1/2 + \epsilon]\Big]\mathbb{P}(X \in [1/2 - \epsilon, 1/2 + \epsilon])$$
$$\leq 2\mathbb{P}(X \in [1/2 - \epsilon, 1/2 + \epsilon])$$
$$= 4\epsilon\Delta \leq \Delta.$$

Since $h_1$ and $h_2$ are identical, we must have that for a sample $\mathcal{S}$ chosen according to either, at least one of $\mathbb{P}(\hat{c}(\mathcal{S}) \geq 1/2) \geq 1/2$ or $\mathbb{P}(\hat{c}(\mathcal{S}) < 1/2) \geq 1/2$. Thus for some $c \in \{c_1, c_2\}$,

$$\mathbb{P}(|\hat{c}(\mathcal{S}) - c| \geq \epsilon) \geq 1/2.$$

$\blacksquare$

# D  Additional Results for IDT with Suboptimal Decision Maker

## D.1  Lower bound for unknown hypothesis class

We give two lower bounds for the sample complexity in the unknown hypothesis class case from Section 4.3. First, in Theorem D.1, we show that there is an IDT problem such that $m = \Omega(\frac{\log(1/\delta)}{p_c\epsilon^2})$ samples are required to estimate $c$. Second, in Theorem D.2, we show that there is an IDT problem such that $m = \Omega(\frac{\sqrt{d}}{p_c\epsilon})$ samples are required. These lower bounds do not precisely match our upper bound of $m = O(\frac{d}{p_c\epsilon^2} + \frac{\log(1/\delta)}{p_c\epsilon^2})$ from Theorem 4.10, and we leave as an open problem the exact minimax sample complexity of IDT in the unknown hypothesis class case. However, they do show that IDT does become harder as the VC-dimension $d$ increases, and that in some suboptimal cases a number of samples proportional to $1/\epsilon^2$ is needed to estimate $c$ to precision $\epsilon$—more than the $1/\epsilon$ needed for an optimal decision maker.

**Theorem D.1 (First lower bound for suboptimal decision maker).** *Fix $0 < \epsilon \le 1/8, 0 < \delta \le 1/2$, and $p_c \le 1/10$. Then there is a decision problem $(\mathcal{D}, c)$, hypothesis class family $\mathbb{H}$, and hypothesis class $\mathcal{H} \in \mathbb{H}$ satisfying the conditions of Theorem 4.10 with the above parameters such that*

$$m < \Omega\left(\frac{\log(1/\delta)}{p_c\epsilon^2}\right) \quad \text{implies that} \quad \mathbb{P}(|\hat{c}(\mathcal{S}) - c| \ge \epsilon) \ge \delta.$$

*Proof.* Specifically, let the sample size

$$m = \frac{\log(1/(2\delta))}{40p_c\epsilon^2}.$$

**Defining the distribution**  First, we define a joint distribution $\mathcal{D}$ over $X = (X_1, X_2) \in \mathcal{X} = \mathbb{R}^2$ and $Y \in \{0, 1\}$. The distribution of $X$ has support on 2 line segments in $\mathbb{R}^2$ and at a point. It can be summarized as follows:

1. $\mathcal{D}_X$ has density $\frac{5p_c}{2}$ on the line segment from $(-1, 0)$ to $(1, 0)$.
   $\mathbb{P}(Y = 1 \mid X = (x_1, 0)) = \frac{1+x_1}{2}$.

2. $\mathcal{D}_X$ has density $10p_cx_1$ at points $(x_1, 1)$ on the line segment from $(0, 1)$ to $(1, 1)$.
   $\mathbb{P}(Y = 1 \mid X = (x_1, 1)) = 1$.

3. $\mathcal{D}_X$ has point mass $\mathbb{P}(X = (-1, 0)) = 1 - 10p_c$.
   $\mathbb{P}(Y = 1 \mid X = (-1, 0)) = 0$.

**Defining the family of hypothesis classes**  Now, we define a family of two hypothesis classes:

$$\mathcal{H}^1 \triangleq \{h(x) = \mathbb{1}\{x_1 \ge b\} \mid b \in [3/8, 5/8]\}$$
$$\mathcal{H}^2 \triangleq \{h(x) = \mathbb{1}\{x_1 \ge b + 2\epsilon x_2\} \mid b \in [1/2, 3/4]\}$$
$$\mathbb{H} \triangleq \{\mathcal{H}^1, \mathcal{H}^2\}.$$

Let's analyze $\mathcal{H}^1$ first. The posterior probability that $Y = 1$ given that $X_1 = x_1$ is

$$\mathbb{P}(Y = 1 \mid X_1 = x_1) = \begin{cases} \frac{1+x_1}{2} & x_1 < 0 \\ \frac{1+9x_1}{2+8x_1} & x_1 \ge 0. \end{cases} \tag{16}$$

It is simple to show that this is increasing in $x_1$; thus, the Bayes optimal decision rule based on $X_1$ for $c$ is

$$h_c^1(x) = \begin{cases} \mathbb{1}\{x_1 \ge 2c - 1\} & c \le 1/2 \\ \mathbb{1}\{x_1 \ge \frac{2c-1}{9-8c}\} & c > 1/2. \end{cases} \tag{17}$$

Now, let's analyze $\mathcal{H}^2$. The posterior probability that $Y = 1$ given that $X_1 - 2\epsilon X_2 = b$ for $b \ge -2\epsilon$ is

$$\mathbb{P}(Y = 1 \mid X_1 - 2\epsilon X_2 = b) = \frac{1 + 9b + 16\epsilon}{2 + 8b + 16\epsilon}. \tag{18}$$

This can also be shown to be increasing in $b$, so the Bayes optimal decision rule based on $X_1 - 2\epsilon X_2$ for $c >= 1/2$ is

$$h_c^2(x) = \mathbb{1}\left\{x_1 - 2\epsilon x_2 \geq \frac{2c - 1 - 16\epsilon + 16c\epsilon}{9 - 8c}\right\}. \tag{19}$$

For this proof, we consider two hypothesis class and loss parameter pairs: $c_1 = 1/2$ for $\mathcal{H}^1$ and $c_2 = \frac{1+16\epsilon}{2+16\epsilon}$ for $\mathcal{H}^2$. These correspond to the decision rules

$$h^1(x) = \mathbb{1}\{x_1 \geq 0\},$$

$$h^2(x) = \mathbb{1}\{x_1 - 2\epsilon x_2 \geq 0\} = \begin{cases} x_1 \geq 0 & x_2 = 0 \\ x_1 \geq 2\epsilon & x_2 = 1. \end{cases}$$

It should be clear that these decision rules agree except when $x_2 = 1$ and $x_1 \in [0, 2\epsilon)$.

Another important fact is that

$$c_2 = \frac{1 + 16\epsilon}{2 + 16\epsilon} = \frac{1}{2} + \frac{4\epsilon}{1 + 8\epsilon} \geq \frac{1}{2} + 2\epsilon \tag{20}$$

since $\epsilon \leq 1/8$.

We defer to the end of the proof to show that these hypotheses and distribution satisfy the conditions of Theorem 4.10.

**Deriving the lower bound** Similarly to the proof of Theorem 4.11, we can bound the probability of an event $E$:

$$\mathbb{P}(\underbrace{\not\exists x_i \in \mathcal{S} \quad x_{i,1} \in [0, 2\epsilon) \wedge x_{i,2} = 1}_{E}) = [1 - \mathbb{P}(X_1 \in [0, 2\epsilon) \wedge X_2 = 1)]^m$$

$$= (1 - 20p_c\epsilon^2)^m$$

$$\geq \left(e^{-40p_c\epsilon^2}\right)^m$$

$$= e^{-\log(1/2\delta)} = 2\delta.$$

Conditional on $E$, the distributions of samples $\mathcal{S}_1$ and $\mathcal{S}_2$ for decision rules $h^1$ and $h^2$ are identical:

$$p_1 = \mathbb{P}(\hat{c}(\mathcal{S}_1) \leq 1/2 + \epsilon \mid E) = \mathbb{P}(\hat{c}(\mathcal{S}_2) \leq 1/2 + \epsilon \mid E),$$
$$p_2 = \mathbb{P}(\hat{c}(\mathcal{S}_1) > 1/2 + \epsilon \mid E) = \mathbb{P}(\hat{c}(\mathcal{S}_2) > 1/2 + \epsilon \mid E).$$

Since $p_1 + p_2 = 1$, at least one of $p_1, p_2 \geq 1/2$. Suppose WLOG that $p_1 \geq 1/2$. Then

$$\mathbb{P}(|\hat{c}(\mathcal{S}_2) - c_2| \geq \epsilon) \overset{(i)}{\geq} \mathbb{P}(\hat{c}(\mathcal{S}_2) \leq 1/2 + \epsilon)$$
$$= \mathbb{P}(\hat{c}(\mathcal{S}_2) \leq 1/2 + \epsilon \mid E)\,\mathbb{P}(E)$$
$$\geq 1/2(2\delta) = \delta.$$

(i) uses the fact shown earlier in (20). Thus, there is a decision problem $(\mathcal{D}, c_2)$ for which the IDT algorithm $\hat{c}$ must make an error of at least size $\epsilon$ with at least probability $\delta$. This concludes the main proof.

**Verifying the requirements of Theorem 4.10** First, we need to show that $q_{\mathcal{H}^1}(X)$ has density at least $p_c$ on $[c_1 - \epsilon, c_1 + \epsilon] = [1/2 - \epsilon, 1/2 + \epsilon]$. From (16) and (17), it is clear that

$$q_{\mathcal{H}^1}(x) = g_1(x_1) = \begin{cases} \frac{1+x_1}{2} & x_1 < 0 \\ \frac{1+9x_1}{2+8x_1} & x_1 \geq 0. \end{cases}$$

We can write the density of $q_{\mathcal{H}^1}(X)$ as the density of $X_1$ multiplied by the derivative of the inverse of $g_1$:

$$p(x_1)\frac{d}{dc}g_1^{-1}(c) \geq \frac{5p_c}{2}\frac{d}{dc}\begin{cases} 2c - 1 & c \leq 1/2 \\ \frac{2c-1}{9-8c} & c > 1/2 \end{cases}$$

$$= \frac{5p_c}{2} \begin{cases} 2 & c \le 1/2 \\ \frac{10}{(9-8c)^2} & c > 1/2 \end{cases}$$

$$\ge p_c.$$

Next, we need to show that $q_{\mathcal{H}^2}(X)$ has density at least $p_c$ on $[c_2 - \epsilon, c_2 + \epsilon] \subseteq [1/2, 1]]$. From (18) and (19), we know that

$$q_{\mathcal{H}^2}(x) = g_2(x_1 - 2\epsilon x_2) = \frac{1 + 9(x_1 - 2\epsilon x_2) + 16\epsilon}{2 + 8(x_1 - 2\epsilon x_2) + 16\epsilon}.$$

Using the same method as for $q_{\mathcal{H}^1}(X)$ and the fact that the density of $X_1 - 2\epsilon X_2$ is at least the density of $X_1$ (i.e., $\frac{5p_c}{2}$), we have that the density of $q_{\mathcal{H}^2}(X)$ is at least

$$\frac{5p_c}{2} \frac{d}{dc} g_2^{-1}(c) = \frac{5p_c}{2} \frac{d}{dc} \frac{2c - 1 - 16\epsilon + 16c\epsilon}{9 - 8c}$$

$$= \frac{5p_c}{2} \frac{10 + 16\epsilon}{(9 - 8c)^2}$$

$$\ge \frac{5p_c}{2} \frac{2}{5} = p_c.$$

The only remaining condition of Theorem 4.10 to prove is MD-smoothness. Again, consider $\mathcal{H}^1$ first:

$$\text{MD}(h_{b_1}^1, \mathcal{H}^2) = \min_{b_2 \in [1/2, 3/4]} \mathbb{P}\left(h_{b_1}^1(X) \ne h_{b_2}^2(X)\right)$$

$$= \min_{b_2 \in [1/2, 3/4]} \frac{5p_c}{2} |b_1 - b_2| + 5p_c \left|b_1^2 - (b_2 + 2\epsilon)^2\right|$$

$$= \frac{5p_c}{2} |b_1 - b_1| + 5p_c \left|b_1^2 - (b_1 + 2\epsilon)^2\right|$$

$$= 20p_c |\epsilon(b_1 + \epsilon)|.$$

From (17), we know that $b_1 - b_1' \le 10(c_1 - c_1')$ where $b_1$ and $b_1'$ are the optimal thresholds for loss parameters $c_1$ and $c_1'$, respectively. So we have that

$$\text{MD}(h_{c_1'}^1, \mathcal{H}^2) - \text{MD}(h_{c_1}^1, \mathcal{H}^2) = 20p_c\epsilon(|b_1' + \epsilon| - |b_1 + \epsilon|)$$

$$\le 20p_c\epsilon|b_1' - b_1|$$

$$\le 200p_c\epsilon|c_1' - c_1|.$$

Thus $h^1$ and $\mathbb{H}$ are $\alpha$-MD-smooth with $\alpha = 200p_c\epsilon$.

Similarly, for $\mathcal{H}^2$,

$$\text{MD}(h_{b_2}^2, \mathcal{H}^1) = \min_{b_1 \in [1/2, 3/4]} \mathbb{P}\left(h_{b_1}^1(X) \ne h_{b_2}^2(X)\right)$$

$$= \min_{b_1 \in [1/2, 3/4]} \frac{5p_c}{2} |b_1 - b_2| + 5p_c \left|b_1^2 - (b_2 + 2\epsilon)^2\right|$$

$$= \frac{5p_c}{2} |b_2 - b_2| + 5p_c \left|b_2^2 - (b_2 + 2\epsilon)^2\right|$$

$$= 20p_c |\epsilon(b_2 + \epsilon)|.$$

So we have that

$$\text{MD}(h_{c_2'}^2, \mathcal{H}^1) - \text{MD}(h^1, \mathcal{H}^1) = 20p_c\epsilon(|b_2' + \epsilon| - |b_2 + \epsilon|)$$

$$\le 20p_c\epsilon|b_2' - b_2|$$

$$\le 200p_c\epsilon|c_2' - c_2|,$$

and thus $h^2$ and $\mathbb{H}$ are also $200p_c\epsilon$-MD-smooth.

∎

**Theorem D.2 (Second lower bound for suboptimal decision maker).** *Let $d \geq 6$ such that $d \equiv 2$ (mod 4). Let $\epsilon \in (0, \frac{1}{64\sqrt{d-2}}]$ and $p_c \in (0, 1]$. Then for any IDT algorithm $\hat{c}(\cdot)$, there is a decision problem $(\mathcal{D}, c)$, hypothesis class family $\mathbb{H}$, and hypothesis class $\mathcal{H} \in \mathbb{H}$ satisfying the conditions of Theorem 4.10 with the above parameters such that*

$$m < \Omega\left(\frac{\sqrt{d}}{p_c \epsilon}\right) \quad \text{implies that} \quad \mathbb{P}(|\hat{c}(\mathcal{S}) - c| \geq \epsilon) \geq \frac{1}{160}.$$

*Proof.* Specifically, let

$$m = \frac{\sqrt{d-2}}{64 p_c \epsilon}.$$

**Defining the distribution**    Let $n = d - 2 \geq 1$; $n$ is divisible by four. First, we define a joint distribution $\mathcal{D}$ over $X \in \mathcal{X} = \mathbb{R}^{n+1}$ and $Y \in \{0, 1\}$. Let $X_j$ refer to the $j$th coordinate of the random vector $X$ and let $x_{ij}$ refer to the $j$th coordinate of the $i$th sample $x_i$. Furthermore, let $X_{1:n}$ refer to the first $n$ components of $X$.

The distribution of $X$ has support on $n$ line segments in $\mathbb{R}^{n+1}$ and at the origin. In particular, it has density $p_c/n$ on each line segment from $(0, \ldots, X_j = 1, \ldots, 0, 0)$ to $(0, \ldots, X_j = 1, \ldots, 0, 1)$, where the density is with respect to the Lebesgue measure on the line. There is additionally a point mass of probability $1 - p_c$ at the origin. Everywhere on the support of $\mathcal{D}$,

$$\mathbb{P}(Y = 1 \mid X_{1:n} = x_{1:n}, X_{n+1} = x_{n+1}) = x_{n+1}.$$

**Defining the family of hypothesis classes**    Next, we define a family of hypothesis classes. Let $\sigma \in \{-1, 1\}^n$ and define

$$f^\sigma(x) = x_{n+1} - 8\epsilon\sqrt{n}\sigma^\top x_{1:n}.$$

Then we define $2^n$ hypothesis classes, one for each value of $\sigma$:

$$\mathcal{H}^\sigma \triangleq \{h(x) = \mathbb{1}\{f^\sigma(x) \geq b\} \mid b \in [1/4, 3/4]\},$$
$$\mathbb{H} \triangleq \{\mathcal{H}^\sigma \mid \sigma \in \{0, 1\}^n\}.$$

Now, we can derive the optimal decision rule in hypothesis class $\mathcal{H}^\sigma$ for loss parameter $c$. Let $[f^\sigma(X)]_{1/4}^{3/4} = \max(1/4, \min(3/4, f^\sigma(X)))$ denote the value $f^\sigma(X)$ clamped to the interval $[1/4, 3/4]$. Then for $b \in (1/4, 3/4)$,

$$\mathbb{P}\left(Y = 1 \mid [f^\sigma(X)]_{1/4}^{3/4} = b\right) = \mathbb{P}\left(Y = 1 \mid X_{n+1} - 8\epsilon\sqrt{n}\sigma^\top X_{1:n} = b\right)$$
$$= \frac{1}{n}\sum_{j=1}^{n}\mathbb{P}\left(Y = 1 \mid X_j = 1 \wedge X_{n+1} = b + 8\epsilon\sqrt{n}\sigma_j\right)$$
$$= b + 8\epsilon\sqrt{n}\frac{\mathbf{1}^\top\sigma}{n}.$$

where $\mathbf{1}$ is the all-ones vector. Thus, the Bayes optimal decision rule based on $[f^\sigma(X)]_{1/4}^{3/4}$ is

$$h_c^\sigma(x) = \mathbb{1}\left\{f^\sigma(x) + 8\epsilon\sqrt{n}\frac{\mathbf{1}^\top\sigma}{n} \geq c\right\}$$
$$= \mathbb{1}\left\{f^\sigma(x) \geq c - 8\epsilon\sqrt{n}\frac{\mathbf{1}^\top\sigma}{n}\right\}$$

for $c - 8\epsilon\sqrt{n}\frac{\mathbf{1}^\top\sigma}{n} \in (1/4, 3/4)$. The induced posterior probability for $\mathcal{H}^\sigma$ is

$$q_{\mathcal{H}^\sigma}(x) = f^\sigma(x) + 8\epsilon\sqrt{n}\frac{\mathbf{1}^\top\sigma}{n}.$$

We consider one hypothesis from each hypothesis class $\mathcal{H}^\sigma \in \mathbb{H}$. Specifically, we consider the optimal decision rule for

$$c^\sigma = \frac{1}{2} + 8\epsilon\sqrt{n}\frac{\mathbf{1}^\top\sigma}{n},$$

which, as shown above is,

$$h^\sigma(x) = \mathbb{1}\left\{f^\sigma(x) \geq \frac{1}{2}\right\}. \tag{21}$$

We leave until the end of the proof to show that each of these decision rules $h_\sigma$ for $\sigma \in \{-1, 1\}^n$ satisfies the requirements of Theorem 4.10.

**Deriving the lower bound** Now, we are ready to derive the lower bound that there is some $h^\sigma$ such that $\mathbb{P}(|\hat{c}(\mathcal{S}) - c| \geq \epsilon) \geq \frac{1}{80}$. First, we can rewrite $h^\sigma$ from (21) as

$$h^\sigma((0, x_j = 1, 0, x_{n+1})) = \mathbb{1}\{x_{n+1} - 8\epsilon\sqrt{n}\sigma_j \geq 1/2\}$$
$$= \mathbb{1}\{x_{n+1} \geq 1/2 + 8\epsilon\sqrt{n}\sigma_j\}.$$

Thus, only decisions made on points where $x_{n+1} \in [1/2 - 8\epsilon\sqrt{n}, 1/2 + 8\epsilon\sqrt{n}]$ are dependent on $\sigma_j$. Denote by $E_j$ the event that there is an observed sample that depends on $\sigma_j$:

$$E_j \quad \triangleq \quad \exists x_i \in \mathcal{S} \text{ such that } x_{ij} = 1 \wedge x_{i,n+1} \in [1/2 - 8\epsilon\sqrt{n}, 1/2 + 8\epsilon\sqrt{n}].$$

Suppose we let $\sigma_j$ be independently Rademacher distributed, i.e. we assign equal probability $1/2^n$ to each $\sigma \in \{-1, 1\}$. Then if $E_j$ does not occur, the sample of decisions $\mathcal{S}$ is independent from $\sigma_j$, i.e.

$$\mathcal{S} \perp\!\!\!\perp \sigma_j \mid \neg E_j.$$

Now let $F$ denote the event that more than $n/2$ of the $E_j$ events occur:

$$F \quad \triangleq \quad |\{j \in 1, \ldots, n \mid E_j\}| > n/2.$$

We will start by proving a lower bound on $\mathbb{P}(|\hat{c}(\mathcal{S}) - c^\sigma| \geq \epsilon \mid \neg F)$. If $F$ does not occur, then at least half of the $E_j$ do not occur. Thus at least half of the elements of $\sigma$ are independent from the sample $\mathcal{S}$. Let $I$ be the set of indices $j$ for which $E_j$ does not occur; thus, $\sigma_I \perp\!\!\!\perp \mathcal{S}$, and given $\neg F$, $|I| \geq n/2$.

We can decompose $c^\sigma$ into part that depends on $\sigma_I$ and part that depends on $\sigma_{I^C}$:

$$c^\sigma = \frac{1}{2} + 8\epsilon\sqrt{n}\frac{\mathbf{1}^\top\sigma_I}{n} + 8\epsilon\sqrt{n}\frac{\mathbf{1}^\top\sigma_{I^C}}{n}. \tag{22}$$

Note that for each $j \in I$, $\frac{\sigma_j+1}{2}$ is $1/2$-Bernoulli distributed. Thus

$$Z = \frac{\mathbf{1}^\top\sigma_I + |I|}{2} = \sum_{j \in I}\frac{\sigma_j + 1}{2} \sim \text{Binom}\left(|I|, \frac{1}{2}\right).$$

We can establish lower bounds on the tails of this given that $F$ occurs:

$$\mathbb{P}\left(Z - \frac{|I|}{2} \geq t \mid \neg F\right) = \left(Z - \frac{|I|}{2} \leq -t \mid \neg F\right) \geq \frac{1}{15}e^{-32t^2/n}.$$

This lower bound is from Matoušek and Vondrák [42]. Plugging in $t = \frac{1}{8}\sqrt{n}$, we obtain

$$\mathbb{P}\left(Z - \frac{|I|}{2} \geq \frac{1}{8}\sqrt{n} \mid \neg F\right) = \left(Z - \frac{|I|}{2} \leq -\frac{1}{8}\sqrt{n} \mid \neg F\right) \geq \frac{1}{20}$$
$$\mathbb{P}\left(\mathbf{1}^\top\sigma_I \geq \frac{1}{4}\sqrt{n} \mid \neg F\right) = \left(\mathbf{1}^\top\sigma_I \leq -\frac{1}{4}\sqrt{n} \mid \neg F\right) \geq \frac{1}{20}. \tag{23}$$

Given $\mathcal{S}$, $\sigma_{I^C}$ is completely known (since $E_j$ occurs for each $j \in I^C$, revealing $\sigma_j$). So plugging (23) into (22) gives

$$\mathbb{P}\left(c^\sigma - \frac{1}{2} - 8\epsilon\sqrt{n}\frac{\mathbf{1}^\top\sigma_{I^C}}{n} \geq 2\epsilon \mid \neg F, \mathcal{S}\right) = \mathbb{P}\left(c^\sigma - \frac{1}{2} - 8\epsilon\sqrt{n}\frac{\mathbf{1}^\top\sigma_{I^C}}{n} \leq -2\epsilon \mid \neg F, \mathcal{S}\right) \geq \frac{1}{20}$$
$$\mathbb{P}\left(c^\sigma - c^{\sigma_{I^C}} \geq 2\epsilon \mid \neg F, \mathcal{S}\right) = \mathbb{P}\left(c^\sigma - c^{\sigma_{I^C}} \leq -2\epsilon \mid \neg F, \mathcal{S}\right) \geq \frac{1}{20}.$$

That is, there is at least probability $1/20$ that $c^\sigma$ is more than $2\epsilon$ above and below $c^{\sigma_{IC}}$, given $\neg F$ and the observed sample $\mathcal{S}$.

This is enough to show that $\mathbb{P}(|\hat{c}(\mathcal{S}) - c^\sigma| \geq \epsilon \mid \neg F, \mathcal{S}) \geq \frac{1}{40}$. First, observe that

$$\mathbb{P}(\hat{c}(\mathcal{S}) \geq c^{\sigma_{IC}} \mid \neg F, \mathcal{S}) + \mathbb{P}(\hat{c}(\mathcal{S}) < c^{\sigma_{IC}} \mid \neg F, \mathcal{S}) = 1,$$

so one of these probabilities must be at least $1/2$. Say WLOG that it is the first. Then

$$\begin{aligned}
\mathbb{P}(|\hat{c}(\mathcal{S}) &- c^\sigma| \geq \epsilon \mid \neg F, \mathcal{S}) \\
&\geq \mathbb{P}(c^\sigma - c^{\sigma_{IC}} \leq -2\epsilon \wedge \hat{c}(\mathcal{S}) \geq c^{\sigma_{IC}} \mid \neg F, \mathcal{S}) \\
&\overset{(i)}{=} \mathbb{P}(c^\sigma - c^{\sigma_{IC}} \leq -2\epsilon \mid \neg F, \mathcal{S}) \, \mathbb{P}(\hat{c}(\mathcal{S}) \geq c^{\sigma_{IC}} \mid \neg F, \mathcal{S}) \\
&\geq \left(\frac{1}{20}\right)\left(\frac{1}{2}\right) = \frac{1}{40}.
\end{aligned}$$

Here, (i) makes use of the fact that $\mathcal{S} \perp\!\!\!\perp \sigma_I \mid \neg F$. Given this, we can finally derive the lower bound on the unconditional probability that $\mathbb{P}(|\hat{c}(\mathcal{S}) - c^\sigma| \geq \epsilon)$:

$$\begin{aligned}
\mathbb{P}(|\hat{c}(\mathcal{S}) &- c^\sigma| \geq \epsilon) \\
&= \mathbb{P}(|\hat{c}(\mathcal{S}) - c^\sigma| \geq \epsilon \mid F)\mathbb{P}(F) + \mathbb{P}(|\hat{c}(\mathcal{S}) - c^\sigma| \geq \epsilon \mid \neg F)\mathbb{P}(\neg F) \\
&\geq \mathbb{P}(|\hat{c}(\mathcal{S}) - c^\sigma| \geq \epsilon \mid \neg F)\mathbb{P}(\neg F) \\
&\geq \frac{\mathbb{P}(\neg F)}{40}.
\end{aligned} \tag{24}$$

So we need to derive a lower bound on $\mathbb{P}(\neg F)$. We can do so by noting that in order for $F$ to occur, there must be at least $n/2$ samples $x_i$ with $x_{i,n+1} \in [1/2 - 8\epsilon\sqrt{n}, 1/2 + 8\epsilon\sqrt{n}]$. The probability of this event for a particular sample is

$$\mathbb{P}\left(X_{n+1} \in [1/2 - 8\epsilon\sqrt{n}, 1/2 + 8\epsilon\sqrt{n}]\right) = 16p_c\epsilon\sqrt{n}.$$

So at least $n/2$ of the $m$ samples must have the event with probability $16p_c\epsilon\sqrt{n}$ occur for $F$ to occur. Let $\text{GE}(p, m, r)$ denote the probability of at least $r$ successes of probability $p$ in $m$ independent trials. Then there is the following fact from probability theory [43]:

$$\text{GE}(p, m, (1 + \gamma)mp) \leq e^{-\gamma^2 mp/3}.$$

Then

$$\begin{aligned}
\mathbb{P}(F) &\leq \text{GE}(16p_c\epsilon\sqrt{n}, m, n/2) \\
&= \text{GE}\left(16p_c\epsilon\sqrt{n}, \frac{\sqrt{n}}{64p_c\epsilon}, 2\left(\frac{\sqrt{n}}{64p_c\epsilon}\right)(16p_c\epsilon\sqrt{n})\right) \\
&\leq e^{-n/12} \leq 3/4
\end{aligned}$$

as long as $n \geq 4$ as assumed. Thus $\mathbb{P}(\neg F) > 1/4$. So putting this together with (24), we have

$$\mathbb{P}(|\hat{c}(\mathcal{S}) - c^\sigma| \geq \epsilon) \geq \frac{1}{160}.$$

This equation is given with respect to the uniform distribution over $\sigma$. But there also must be a particular $\sigma$ and thus corresponding $h^\sigma \in \mathcal{H}^\sigma$ which has the same tails on $\hat{c}(\mathcal{S}) - c$. Thus we conclude the proof.

**Verifying the requirements of Theorem 4.10**    Now we show that the distribution and hypothesis class family satisfy the conditions of Theorem 4.10. First, note that all $h \in \mathcal{H} \in \mathbb{H}$ are thresholds on linear functions of the observation $x$. Thus, $\cup_{\mathcal{H} \in \mathbb{H}}\mathcal{H}$ is a subset of the halfspaces in $\mathbb{R}^{n+1}$ and so it has VC-dimension at most $n + 2 = d$.

Next, it is clear that for $\rho \leq \epsilon$,

$$\mathbb{P}(q_{\mathcal{H}^\sigma}(X) \in (c, c + \rho]) = \mathbb{P}\left(f^\sigma(X) + 8\epsilon\sqrt{n}\frac{\mathbf{1}^\top\sigma}{n} \in (c, c + \rho]\right)$$

$$= \sum_{j=1}^{n} \mathbb{P}\left(X_j = 1 \wedge X_{n+1} - 8\epsilon\sqrt{n}\sigma_j + 8\epsilon\sqrt{n}\frac{\mathbf{1}^\top \sigma}{n} \in (c, c+\rho]\right)$$

$$= \sum_{j=1}^{n} \frac{p_c \rho}{n} = p_c \rho.$$

A similar result can be shown for $\mathbb{P}(q_{\mathcal{H}^\sigma}(X) \in [c - \rho, c))$.

Finally, we need to show that MD-smoothness holds. Take any $h^\sigma$ and any $\mathcal{H}^{\tilde\sigma}$. Then the disagreement between $h^\sigma$ and a hypothesis in $\mathcal{H}^{\tilde\sigma}$ with threshold $b$ is

$$\mathbb{P}(h^\sigma(X) \neq h_b^{\tilde\sigma}(X)) = \frac{p_c}{n} \sum_{j=1}^{n} \left| \frac{1}{2} + 8\epsilon\sqrt{n}\sigma_j - b - 8\epsilon\sqrt{n}\tilde\sigma_j \right|$$

$$= \frac{p_c}{n} \sum_{j=1}^{n} \left| \left( \frac{1}{2} + 8\epsilon\sqrt{n}(\sigma_j - \tilde\sigma_j) \right) - b \right|.$$

This is minimized when $b$ is the median of $\left( \frac{1}{2} + 8\epsilon\sqrt{n}(\sigma_j - \tilde\sigma_j) \right)$ for $j = 1, \ldots, n$. Thus $b \in [\frac{1}{2} - 8\epsilon\sqrt{n}, \frac{1}{2} + 8\epsilon\sqrt{n}]$; since $\epsilon \leq \frac{1}{64\sqrt{n}}$, this implies $b \in [3/8, 5/8]$. Suppose now we let $c' \in [c^\sigma - 1/8, c^\sigma + 1/8]$. Then we can let $b' = b + (c' - c^\sigma)$ and

$$\text{MD}(h_{c'}^\sigma, \mathcal{H}^{\tilde\sigma}) \leq \mathbb{P}\left( h_{c'}^\sigma(X) \neq h_{b'}^{\tilde\sigma}(X) \right) = \mathbb{P}\left( h^\sigma(X) \neq h_b^{\tilde\sigma}(X) \right) = \text{MD}(h^\sigma, \mathcal{H}^{\tilde\sigma}).$$

Thus for $|c' - c^\sigma| \leq 1/8$, $h^\sigma$ and $\mathbb{H}$ are 0-MD-smooth. If $|c' - c^\sigma| > 1/8$, then we have

$$\text{MD}(h_{c'}^\sigma, h^{\tilde\sigma}) \leq 1 < \frac{8}{\text{MD}(h^\sigma, \mathcal{H}^{\tilde\sigma})}|c' - c^\sigma|\text{MD}(h^\sigma, \mathcal{H}^{\tilde\sigma}).$$

Thus overall $h^\sigma$ and $\mathbb{H}$ are $\alpha$-MD-smooth with

$$\alpha = \max_{\tilde\sigma \neq \sigma} \frac{8}{\text{MD}(h^\sigma, \mathcal{H}^{\tilde\sigma})}.$$

∎

*Bibliographic note:* we establish dependence on the VC dimension $d$ in Theorem D.2 using a technique similar to that used by Ehrenfeucht et al. [44].

## D.2 Necessity of MD-smoothness

The lower bounds given in Section D.1 do not depend on the $\alpha$ parameter from the MD-smoothness assumption made in Theorem 4.3; thus, one may wonder if this assumption is necessary. In the following lemma, we show that it is necessary in some cases by giving an example of an IDT problem where a lack of MD-smoothness precludes identifiability of the loss parameter.

**Lemma D.3 (No MD-smoothness can prevent identifiablity).** *Let $\epsilon \in (0, 1/10)$. Then for any IDT algorithm $\hat{c}(\cdot)$, there is a decision problem $(\mathcal{D}, c)$, hypothesis class family $\mathbb{H}$, and hypothesis class $\mathcal{H} \in \mathbb{H}$ satisfying the conditions of Theorem 4.10 except for MD-smoothness such that*

$$\mathbb{P}(|\hat{c}(\mathcal{S}) - c| \geq \epsilon) \geq \frac{1}{2}$$

*for a sample $\mathcal{S}$ of any size $m$.*

*Proof.* **Defining the distribution** First, we define a distribution $\mathcal{D}$ over $X \in \mathcal{X} = \mathbb{R}^2$ and $Y \in \{0, 1\}$. $\mathcal{D}_X$ has density $1/2$ on two squares $[-1, 0] \times [-1, 0]$ and $[0, 1] \times [0, 1]$, and the distribution of $Y \mid X$ is defined as follows:

$$\mathbb{P}(Y = 1 \mid X = x = \begin{cases} \frac{2}{3} + \frac{2}{15}x_1 + \frac{8}{15}x_2 & x \in [-1, 0] \times [-1, 0] \\ \frac{1}{3} + \frac{8}{15}x_1 + \frac{2}{15}x_2 & x \in [0, 1] \times [0, 1]. \end{cases}$$

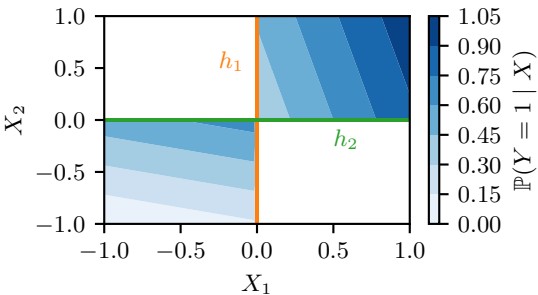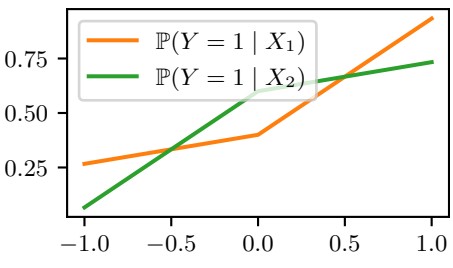

Figure 6: A visualization of the distribution and decision rules used in Lemma D.3 to show that a lack of MD-smoothness can prevent identifiability of the loss parameter $c$. On the left, the distribution over $X = (X_1, X_2)$ and $Y$ is shown; $X$ has constant density on unit squares in the first and third quadrants, and $\mathbb{P}(Y = 1 \mid X)$ varies as shown with the heatmap. We consider two decision rules $h_1$ and $h_2$ which are optimal thresholds of $X_1$ and $X_2$, respectively, for loss parameters $c_1 = 2/5$ and $c_2 = 3/5$, respectively. Since $c_1 \neq c_2$ but $\mathbb{P}(h_1(X) = h_2(X)) = 1$, it is impossible to identify $c$ reliably. This is because the distribution and decision rules are not MD-smooth, since shifting either decision rule slightly causes a jump in minimum disagreement with the other hypothesis class from 0 to a positive value.

**Defining the family of hypothesis classes**  We consider the two hypothesis classes which are thresholds on one component of the observation $x$:

$$\mathcal{H}_1 = \{h(x) = \mathbb{1}\{x_1 \geq b\} \mid b \in [-1, 1]\},$$
$$\mathcal{H}_2 = \{h(x) = \mathbb{1}\{x_2 \geq b\} \mid b \in [-1, 1]\}.$$

That is, $\mathbb{H} = \{\mathcal{H}_1, \mathcal{H}_2\}$. The conditional probabilities for $Y = 1$ given just one of the observation components are

$$q_{\mathcal{H}_1}(x) = \mathbb{P}(Y = 1 \mid X_1 = x_1) = \frac{2}{5} + \frac{2}{15}x_1 + \frac{2}{5}x_1\mathbb{1}\{x_1 \geq 0\},$$
$$q_{\mathcal{H}_2}(x) = \mathbb{P}(Y = 1 \mid X_2 = x_2) = \frac{3}{5} + \frac{2}{15}x_2 + \frac{2}{5}x_2\mathbb{1}\{x_2 \leq 0\}. \tag{25}$$

We consider the optimal decision rules for $c_1 = 2/5$ and $c_2 = 3/5$ in $\mathcal{H}_1$ and $\mathcal{H}_2$, respectively, which from the above can be calculated as

$$h_1(x) = \mathbb{1}\{x_1 \geq 0\},$$
$$h_2(x) = \mathbb{1}\{x_2 \geq 0\}.$$

The distribution and decision rules are visualized in Figure 6.

**Lack of identifiability**  Note that since $X$ only has support where $\operatorname{sgn}(X_1) = \operatorname{sgn}(X_2)$, the above decision rules are indistinguishable. Thus, we use the same techniques from Corollary 4.12 and Lemma C.1 to show that for at least one of $c \in \{c_1, c_2\}$

$$\mathbb{P}(|\hat{c}(\mathcal{S}) - c| \geq 1/2(c_2 - c_1) = 1/10 \geq \epsilon) \geq 1/2.$$

**Hypothesis classes are not MD-smooth**  Although this is not required for the proof of the lemma, we will demonstrate that the defined hypothesis classes are not $\alpha$-MD-smooth for any $\alpha$. By way of contradiction, assume that there is some $\alpha$ such that $h_1$ and $\mathbb{H}$ are MD-smooth. Then for any $c_1' \in [0, 1]$,

$$\operatorname{MD}(h_{c_1'}, \mathcal{H}_2) \leq (1 + \alpha|c_1' - c_1|)\operatorname{MD}(h_1, \mathcal{H}_2) = 0.$$

Here, $\operatorname{MD}(h_1, \mathcal{H}_2)$ since $\mathbb{P}(h_1(X) \neq h_2(X)) = 0$, i.e. $h_1$ and $h_2$ do not disagree at all. However, there are clearly values of $c_1'$ such that $\operatorname{MD}(h_{c_1'}, \mathcal{H}_2) > 0$, so we have a contradiction.

**Verifying the other requirements of Theorem 4.10**  Clearly, the family of hypothesis classes defined above have finite VC-dimension.

The densities of $q_{\mathcal{H}_1}(X)$ and $q_{\mathcal{H}_2}(X)$ can be calculated as the density of $X_1$ or $X_2$ multiplied by the derivative of the inverse of the posterior probability functions. The densities of $X_1$ and $X_2$ are

both $1/2$ on the interval $[-1, 1]$, and the derivative of the inverse of the equations in (25) is at least $15/8$. So the distribution satisfies the requirements of Theorem 4.10 other than MD-smoothness with $p_c \geq 15/16$.

■

# E  Feature Subset Hypothesis Class Family

In this section, we work through the application of Theorem 4.10 to a practical example. Theorem 4.10 concerns the case of IDT when the decision maker could be restricting themselves to any suboptimal hypothesis class $\mathcal{H} \in \mathbb{H}$ for some family of hypothesis classes $\mathbb{H}$. In this example, we consider $\mathbb{H}_{\text{feat}}$ as defined in (1) and repeated here:

$$\mathbb{H}_{\text{feat}} \triangleq \{\mathcal{H}_S \mid S \subseteq \{1, \ldots, n\}\} \quad \text{where} \quad \mathcal{H}_S \triangleq \left\{ h(x) = f(x_S) \mid f : \mathbb{R}^{|S|} \to \{0, 1\} \right\}. \quad (1)$$

This family can model decision makers that have bounded computational capacity and may only be able to reason based on a few features of the data. An application of structural risk minimization [45] from learning theory shows that the sample complexity of IDT in this case may scale only linearly in the number of features considered and logarithmically in the total feature count:

**Lemma E.1.** *Let a decision maker use a hypothesis class $\mathcal{H}_S \in \mathbb{H}_{\text{feat}}$ as defined in (1) which consists of decision rules depending only on the subset of the features in $S$. Let $s = |S|$ be the number of such features; neither $s$ nor $S$ is known. Suppose $\mathcal{X} = \mathbb{R}^d$, i.e. $d$ is the total number of features. Let assumptions on $\epsilon$, $\delta$, $\alpha$, and $p_c$ be as in Theorem 4.10.*

*Let $\hat{h}_{\hat{c}} \in \arg\min_{\hat{h} \in \mathcal{H}_{\hat{S}}} \mathcal{R}_{\hat{c}}(\hat{h})$ be chosen to be consistent with the observed decisions, i.e. $\hat{h}_{\hat{c}}(x_i) = \hat{y}_i$, and such that $|\hat{S}|$ is as small as possible. Then $|\hat{c} - c| \leq \epsilon$ with probability at least $1 - \delta$ as long as the number of samples $m$ satisfies*

$$m \geq O\left[\left(\frac{\alpha}{\epsilon} + \frac{1}{\epsilon^2}\right)\left(\frac{s \log d + \log(1/\delta)}{p_c}\right)\right].$$

*Proof.* We prove Lemma E.1 by bounding the VC-dimension of the union of all optimal decision rules in all $\mathcal{H}_S \in \mathbb{H}_{\text{feat}}$ where $|S| \leq s$. An optimal decision rule for loss parameter $c$ in $\mathcal{H}_S$ is given by the Bayes optimal classifier:

$$h_c^S(x) = \mathbb{1}\{\mathbb{P}(Y = 1 \mid X_S = x_s) \geq c\}.$$

Now consider a set of observations $x_1, \ldots, x_d \in \mathcal{X}$. We will show that for $d > 1 + 2s \log_2(n+1)$, this set cannot be shattered by $d$. To see why, note that decision rules in any particular class $\mathcal{H}_S$ threshold the posterior probability $\mathbb{P}(Y = 1 \mid X_S = x_s)$. Thus, each hypothesis class can only produce $d + 1$ distinct labelings of the set of observations. The number of hypothesis classes $\mathcal{H}_S$ with $|S| \leq s$ is

$$\sum_{k=0}^{s} \binom{n}{s} \leq \sum_{k=0}^{s} n^k \leq (n+1)^s.$$

So the number of distinct labelings assigned by hypotheses in $\mathbb{H}$ to the observations must be at most $(d+1)(n+1)^s < 2^d$ if $d > 1 + 2s \log_2(n+1)$. Thus this set cannot be shattered, so

$$\text{VCdim}\left(\cup_{|S| \leq s} \mathcal{H}_S\right) \leq 1 + 2s \log_2(n+1) = O(s \log n).$$

Applying Theorem 4.10 with $d = O(s \log n)$ completes the proof. ■

The following lemma states conditions under which $\alpha$-MD-smoothness holds for $\mathbb{H}_{\text{feat}}$.

**Lemma E.2.** *Let $\mathbb{H}_{\text{feat}}$ and $\mathcal{H}_S$ be defined as in (1). Let $h \in \mathcal{H}_S$. Suppose that there is a $\zeta > 0$ such that for any $\hat{S} \subseteq \{1, \ldots, n\}$, one of the following holds: either (a) $\mathbb{P}(Y = 1 \mid X = x_S) = \mathbb{P}(Y = 1 \mid X = x_{\hat{S}})$ for all $x \in \mathbb{R}^d$, or (b) $MD(h, \mathcal{H}_{\hat{S}}) \geq \zeta$. Furthermore, suppose that the distribution of $q_{\mathcal{H}_S}(X)$ is absolutely continuous with respect to the Lebesque measure and that its density is bounded above by $M < \infty$. Then $h$ and $\mathbb{H}_{\text{feat}}$ are $\alpha$-MD-smooth with $\alpha = M/\zeta$.*

Since $\alpha$-MD-smoothness is a sufficient condition for identification of the loss function parameter $c$, Lemma E.2 gives conditions under which IDT can be performed. The main requirement is that considering different subsets of the features either gives identical decision rules (case (a)) or decision rules which disagree by some minimum amount (case (b)). If decision rules using a different subset of the features can be arbitrarily close to the true one, it may not be possible to apply IDT.

*Proof.* Consider any $\hat{S} \subseteq \{1, \ldots, n\}$. If (a) holds for $\hat{S}$, then $h_c^S(x) = h_c^{\hat{S}}(x)$ for any $c \in [0, 1]$ and $x \in \mathcal{X}$. Thus

$$\mathrm{MD}(h_{c'}^S, \mathcal{H}_{\hat{S}}) = 0 \le (1 + \alpha|c' - c|)\mathrm{MD}(h_c^S, \mathcal{H}_{\hat{S}}) = 0$$

so $\alpha$-MD-smoothness holds in this case for any $\alpha$.

If (b) holds, then let $\hat{h} \in \arg\min_{\hat{h} \in \mathcal{H}_{\hat{S}}} \mathbb{P}(h(X) \ne \hat{h}(X))$. Let $c' \in [0, 1]$; without loss of generality, we may assume that $c' > c$. Denote $q_S(x) = \mathbb{P}(Y = 1 \mid X_S = x_s)$. Then

$$\mathrm{MD}(h_{c'}^S, \mathcal{H}_{\hat{S}})$$

$$\le \mathbb{P}(h_{c'}^S(X) \ne \hat{h}(X))$$

$$= \mathbb{P}\Big(q_S(X) < c' \wedge \hat{h}(X) = 1)\Big) + \mathbb{P}\Big(q_S(X) > c' \wedge \hat{h}(X) = 0)\Big)$$

$$\le \mathbb{P}\Big(q_S(X) \in [c, c') \wedge \hat{h}(X) = 1\Big) + \mathbb{P}\Big(q_S(X) < c \wedge \hat{h}(X) = 1\Big) + \mathbb{P}\Big(q_S(X) > c \wedge \hat{h}(X) = 0)\Big)$$

$$= \mathbb{P}\Big(q_S(X) \in [c, c') \wedge \hat{h}(X) = 1\Big) + \mathrm{MD}(h, \mathcal{H}_{\hat{S}})$$

$$\le M(c' - c) + \mathrm{MD}(h, \mathcal{H}_{\hat{S}})$$

$$\le \left[1 + \frac{M}{\zeta}(c' - c)\right]\mathrm{MD}(h, \mathcal{H}_{\hat{S}}).$$

So $h$ and $\mathbb{H}$ satisfy $\alpha$-MD-smoothness with $\alpha = M/\zeta$. ∎

# F   Surrogate Loss Functions

Here, we explore using IDT when the decision maker minimizes a surrogate loss instead of the true loss. So far, as formulated in Section 3, we have assumed that the decision maker chooses a decision rule $h$ which minimizes the expected loss $\mathbb{E}[\ell_c(h(X), Y)]$, where the loss function is defined as

$$\ell_c(\hat{y}, y) = \begin{cases} 0 & \hat{y} = y \\ c & \hat{y} = 1 \wedge y = 0 \\ 1 - c & \hat{y} = 0 \wedge y = 1 \end{cases}$$

$$= \begin{cases} c\,\mathbb{1}\{\hat{y} = 1\} & y = 0 \\ (1 - c)\,\mathbb{1}\{\hat{y} = 0\} & y = 1. \end{cases} \tag{26}$$

However, this loss function is not convex or continuous, so it is difficult to optimize. Thus, we might expect the decision maker to choose their decision rule using a *surrogate loss* which is convex. In particular, suppose that the decision rule $h(\cdot)$ is calculated by thresholding a function $f : \mathcal{X} \to \mathbb{R}$:

$$h(x) = \mathbb{1}\{f(x) \ge 0\}.$$

Then, we can replace the indicator functions in (26) with a surrogate loss $V : \mathbb{R} \to \mathbb{R}$:

$$\tilde{\ell}_c(w, y) = \begin{cases} c\,V(w) & y = 0 \\ (1 - c)\,V(-w) & y = 1. \end{cases} \tag{27}$$

Say that the decision maker minimizes this loss $\tilde{\ell}_c$ instead of the true loss $\ell$:

$$f^* \in \arg\min_f \mathbb{E}[\tilde{\ell}_c(f(X), Y)]. \tag{28}$$

The following lemma shows that, for reasonable surrogate losses, if the decision maker is optimal then minimizing the surrogate loss is equivalent to minimizing the true loss. The proof is adapted from Section 4.2 of Rosasco et al. [46]; they show that the hinge loss, squared loss, and logistic loss all satisfy the necessary conditions.

**Lemma F.1.** *Suppose $V : \mathbb{R} \to \mathbb{R}$ is convex and that it is strictly increasing in a neighborhood of 0. Let $f^*$ be chosen as in (28), and let $h(x) = \mathbb{1}\{f^*(x) \geq 0\}$. Then $h \in \arg\min_h \mathbb{E}[\ell_c(h(X), Y)]$; that is, the threshold of $f^*$ is an optimal decision rule for the true cost function.*

*Proof.* We prove the lemma by contradiction; assume that $h$ is *not* an optimal decision rule for the true loss function. Then by Lemma 4.1,

$$\mathbb{P}(h(X) \neq \mathbb{1}\{q(X) \geq c\} \wedge q(X) \neq c) > 0.$$

This implies that either

$$\mathbb{P}(h(X) = 0 \wedge q(X) > c) > 0 \qquad \text{or} \qquad \mathbb{P}(h(X) = 1 \wedge q(X) < c) > 0,$$

or equivalently,

$$\mathbb{P}(f^*(X) < 0 \wedge q(X) > c) > 0 \qquad \text{or} \qquad \mathbb{P}(f^*(X) \geq 0 \wedge q(X) < c) > 0. \qquad (29)$$

Without loss of generality, assume the former. Define

$$\tilde{f}(x) = \begin{cases} 0 & f^*(x) < 0 \wedge q(x) > c \\ f^*(x) & \text{otherwise.} \end{cases}$$

Consider any $x$ which satisfies $f^*(x) < 0$ and $q(x) > c$. We can write

$$\mathbb{E}\left[\tilde{\ell}_c(f^*(X), Y) - \tilde{\ell}_c(\tilde{f}(X), Y) \mid X = x\right]$$

$$= \mathbb{P}(Y = 0 \mid X = x)\, c\, \left(V(f^*(x)) - V(\tilde{f}(x))\right) + \mathbb{P}(Y = 1 \mid X = x)\,(1 - c)\,\left(V(-f^*(x)) - V(-\tilde{f}(x))\right)$$

$$= (1 - q(x))\, c\, (V(f^*(x)) - V(0)) + q(x)\,(1 - c)\,(V(-f^*(x)) - V(0))$$

$$= \tilde{\ell}_c(f^*(x) \mid x) - \tilde{\ell}_c(0 \mid x),$$

where we define

$$\tilde{\ell}_c(w \mid x) = (1 - q(x))\, c\, V(w) + q(x)\,(1 - c)\, V(-w).$$

$\tilde{\ell}_c(w \mid x)$ satisfies two properties:

1. It is convex in $w$, since it is a sum of two convex functions.

2. It is strictly decreasing in $w$ in a neighborhood of 0. To see why, note that we assumed $q(x) > c$, so
$$(1 - q(x))\, c < (1 - c)\, c < q(x)\,(1 - c).$$
Thus, since the weight on $V(-w)$ is greater than the weight on $V(w)$, and $V(w)$ is strictly increasing about 0, $\tilde{\ell}_c(w \mid x)$ must be strictly decreasing about 0.

Together, these properties imply that

$$\tilde{\ell}_c(f^*(x) \mid x) - \tilde{\ell}_c(0 \mid x) > 0$$

since we assumed that $f^*(x) < 0$. Thus we have that

$$\mathbb{E}\left[\tilde{\ell}_c(f^*(X), Y) - \tilde{\ell}_c(\tilde{f}(X), Y) \mid X = x\right] > 0 \qquad (30)$$

for any $x$ where $f^*(x) < 0$ and $q(x) > c$.

Now, we analyze the difference in expect loss for $f^*$ and $\tilde{f}$. Since these agree on all points except when $f^*(x) < 0$ and $q(x) > c$, we have that

$$\mathbb{E}[\tilde{\ell}(f^*(X), Y)] - \mathbb{E}[\tilde{\ell}(\tilde{f}(X), Y)]$$

$$= \mathbb{E}\left[\tilde{\ell}(f^*(X), Y) - \tilde{\ell}(\tilde{f}(X), Y) \,\Big|\, f^*(X) < 0 \wedge q(X) > c\right] \mathbb{P}\left(f^*(X) < 0 \wedge q(X) > c\right)$$

$$\overset{(i)}{>} 0. \qquad (31)$$

Here, (i) is due to the combination of (30), which implies the first term is positive, and the first case of (29), which implies the second term is positive.

(31) implies that $\tilde{f}$ has lower expected surrogate loss than $f^*$. However, we assumed that $f^*$ minimized the expected surrogate loss; thus we have a contradiction. ∎

| Setting | True loss | Surrogate loss |
|---|:---:|:---:|
| IDT for optimal decision maker (Theorem 4.2) | ✓ | ✓ |
| IDT for suboptimal decision maker (Theorems 4.7 and 4.10) | ✓ | ✗ |
| No identifiability for decisions without uncertainty (Corollary 4.12) | ✓ | ✓ |

Table 1: An overview of which of our results apply in the setting when the decision maker is minimizing a surrogate loss rather than the true loss.

Lemma F.1 means that all the results for an optimal decision maker (e.g., Theorem 4.2) apply immediately to a decision maker minimizing a reasonable surrogate loss. In the case of decision problems without uncertainty, the decision rule will encounter zero loss and thus must be optimal, so Lemma F.1 also applies in this case for an optimal or suboptimal decision maker (e.g., Corollary 4.12). In the case of a suboptimal decision maker facing uncertainty, different loss functions may lead to different decision rules, so we cannot extend the results in that case to surrogate losses. Table 1 summarizes which results hold equivalently for decision makers minimizing an expected surrogate loss.

## G    Further Comparison to Prior Work

In this section, we compare two prior papers on preference learning to our results. Mindermann et al. [20] and Bıyık et al. [21] both propose methods for active preference learning, i.e. querying a person to learn their preferences. In each method, queries are prioritized which minimize the uncertainty of the person. The authors argue that such queries are easier to answer and thus lead to more effective preference learning. At first, these results may seem to contradict our findings that uncertain decisions make preference learning easier. However, we argue that their results are not in conflict with ours. Decisions with more uncertainty are probably more difficult for people to make, and those close to the decision boundary are probably the most difficult. However, our results show that it is *necessary* to observe such decisions in order to recover the person's preferences. If we cannot observe decisions made arbitrarily close to the person's decision boundary, we cannot exactly characterize the loss function they are optimizing. Thus, combining the results of Mindermann et al. [20] and Bıyık et al. [21] with ours suggests that there is a tradeoff between the ease of the decision problem for the human and the identifiability of their preferences. That is, uncertainty may make the human's decision problem more difficult but our problem of identifying preferences easier.