# OpenReview forum: "Uncertain Decisions Facilitate Better Preference Learning"
_NeurIPS.cc/2021/Conference — NeurIPS 2021 Spotlight_

### Official Review · Reviewer_FDLv · 2021-07-06

**Rating:** 7
**Confidence:** 3

**Summary:**

The authors study the problem of learning the preferences of a human decision-maker from past decisions. The setting is that feature–decision pairs (X, Y) are drawn from a known distribution $\mathcal{D}$. Initially the authors assume the Bayes-optimal case, where the decision-maker perfectly knows P(Y | X), and the objective is to identify the decision boundary separating positive labels from negative labels in terms of P(Y | X) conditional belief. The authors take a PAC approach, give bounds on the number of samples required to identify the boundary with probability $1 - \delta$. They later relax the Bayes-optimal rationality assumption, considering the case of a limited hypothesis class and the case of a family of hypothesis classes.



**Limitations And Societal Impact:**

The authors have a paragraph devoted to limitations that satisfied my concerns. In my opinion, the largest limitation is the restriction to expected utility as a metric. A natural successor of this paper considers risk sensitivity and constraints that the decision must satisfy.

I do not consider negative societal impact to be an issue with this paper.

**Main Review:**

I found this to be a clear, useful paper that tackles a natural problem that builds on the literature. The main line of analysis is fairly straightforward, but the author's approach is quite thorough and goes quite a bit beyond the "minimum" needed to make the main point. This "additional" content is rather useful and really helps to explain their approach and how it could be used (e.g., the counterexample showing the lack of MD-smoothness, the characterization of MD-smoothness for the "ignoring some features" hypothesis class, lower bounds for the problems of interest, and the fairness example). There are several directions of interest that build off the approach taken by the paper.

The risks of the paper that would push me towards rejection are:
- Significant math errors—I couldn't find any, but I did not check thoroughly.
- The ideas and techniques are well known enough that there could have been another paper to do this in the last decade. To the best of my knowledge, no such paper exists, but the risk is larger than normal.

Minor comments: I did not find the introduction clear—the cat vs. pandemic example did not work for me. The authors say that the pandemic quarantine decision is subjective and thus, they seem to imply that the cat decision is objective. I don't think that argument works—if the cat decision is objective, there is nothing to learn about the decision-maker at all. A suggestion would be to use a case where there is a hypothetically perfectly accurate COVID test instead of the cat example. Both decisions are then subjective, but there is no uncertainty in the perfect test case, and there is thus very little we can learn about the preferences of the decision-maker.

Post-response: I am maintaining my score. The authors proposed changes will increase clarity.

**Time Spent Reviewing:**

3.5

---

> ### Author Response · Authors · 2021-08-10
> **Response to Reviewer FDLv**
>
> Thank you for your positive review and the feedback on the clarity of the introduction.
>
> **Regarding the risks you mentioned:** we would like to emphasize that we have thoroughly checked all the proofs, and that intuition, upper and lower bounds, and counterexamples are all consistent with each other. In addition, we conducted an extensive literature search for similar results in past work and were unable to find any. Since the other reviewers did not find significantly similar past work either, we believe the risk that our results duplicate others’ is minimal. We understand the reviewer’s concern because the core idea is clear in hindsight, but it is still important that such ideas are published.
>
> **Regarding the introduction:** we agree that the description of decisions as “subjective” and “objective” is confusing; we will remove these terms. Per your suggestion, we plan to replace the cat vs. pandemic quarantine example with a table of a few similar pairs of decisions, where one has uncertainty and one does not. Here are some we are planning to include:
>
> | No uncertainty                                                                             | Uncertainty                                                                                   |
> |--------------------------------------------------------------------------------------------|-----------------------------------------------------------------------------------------------|
> | Should I call my friend’s pet a dog or a cat?                                              | Should I avoid this animal in the forest that could be either a deer or a bear?               |
> | Should I quarantine a traveler with a 100% accurate negative test for a dangerous disease? | Should I quarantine a traveler with some symptoms of a dangerous disease but no test results? |
> | Should a person with irrefutable evidence of and confession to a crime be convicted?       | Should a person with circumstantial evidence of a crime be convicted?                         |

---

### Official Review · Reviewer_gz3s · 2021-07-12

**Rating:** 6
**Confidence:** 4

**Summary:**

The authors analyze learning preferences from human decisions in one-shot decision problems. They make 4 main contributions: 1) characterize sample complexity in this setting for the goal of identifying the cost function parameter (as opposed to doing prediction) 2) show that sample complexity improves when the decisions are uncertain 3) repeat contributions 1 & 2 when the human is suboptimal and can therefore only choose from a limited set of policies 4) show under what conditions, when the set of policies is unknown, one can still recover the cost function.


**Limitations And Societal Impact:**

Yes

**Main Review:**

In my view, the most important contribution is under what conditions preferences can be identified without knowing how the human makes decisions. This is a thorny and major open problem that limits the prospects for learning human preferences. In this paper, each model of human decision-making is described by a limited hypothesis class that the human can choose from. Identifying the true class is possible with a smoothness assumption on the set of hypothesis classes. However, the authors could better clarify the strength of this assumption.

The findings are not only interesting - it is easy to see they it could be used to develop better preference learning techniques.

The paper has some clarity issues. If they are resolved I may increase my score. For example the introduction talks about human uncertainty but this concept is never defined and doesn’t appear to be used. Instead the theoretical analysis requires uncertainty in the sense that the _agent’s_ posteriors haven’t concentrated (for every decision). Or at least that is my reading of the text. Figure 1 doesn’t help improve this confusion. Please see Houlsby et al (https://arxiv.org/abs/1112.5745) for discussion of these different uncertainties. More details below.

The first contribution, that uncertainty helps with identifying preferences, seems novel in this setting but a similar contribution is made in Mindermann et al. (https://arxiv.org/abs/1809.03060) which points out that cost function parameters can be better identified by excluding the optimal choice from the set of choices for the human. This paper and Biyik et al (https://arxiv.org/abs/1910.04365) also suggest that human uncertainty can hurt preference learning; the contrast to your result should be discussed. Furthermore, many papers have pointed out that decisions only convey information when they are uncertain _under the agent’s current posterior_.


___Detailed comments___

Well-written abstract and introduction

Thm 4.2: how do you account for the fact that the posterior changes after each new sample?

Please explain MD-smoothness and how plausible it is. This is important if you want theorem 4.10 to be a main contribution.

Some definitions are lacking.
- Please define the human policy
- Please define partial observability. What is not observed. You seem to be using it in a non-standard way. It usually means that the state x is hidden but this is not the case here. Indeed I think you might be looking for a different term.
- Please define uncertainty (see above). Uncertainty in what?
- Please define the posterior probability, conditioned on some data. And the distribution over posterior probabilities

The paper has somewhat disparate contributions ranging from decisions under uncertainty to suboptimal decisions. Perhaps these could be separated into two papers.

Typo: “between between”


**Time Spent Reviewing:**

3

---

> ### Author Response · Authors · 2021-08-10
> **Response to Reviewer gz3s**
>
> Thank you for the insightful review and the time spent identifying areas in which we could improve the clarity of the paper. Here are responses to the concerns you raised in your review:
>  * **Clarity on definitions:** we apologize for the confusion around the various definitions of “uncertainty,” “partial observability,” and the beliefs of the human decision maker. Our model is that the decision maker knows the true distribution of observations $X$ and correct decisions (targets) $Y$. Thus, the posterior distribution $\mathbb{P}(Y = 1 | X = x)$ is known to both the decision maker and us, and it does not change while the decision maker is making decisions. That is, the only uncertainty that must be considered while making the decision is  contained in this posterior distribution on $Y$ given the observation $X$. The distribution could have been estimated from past data using some statistical method; we touch on this in the “limitations and future work” section. We do not consider that the human is using active learning as in Houlsby et al, or that they are learning at all. While it would also be interesting to analyze the case when the decision maker is learning as they go (updating their decision rule), we leave this for future work. The optimal decision rule is given by $h(x) = \mathbf{1}\\{\mathbb{P}(Y = 1 | X = x) \geq c\\}$, where $c$ is the loss function parameter introduced in Section 3.
>
>    Given this framework, we define a decision problem with *no uncertainty* as one where $\mathbb{P}(Y = 1 | X = x) \in \\{0, 1\\}$ almost surely. That is, the correct decision or label is always sure given the observation x. We will add this definition to the paper; thank you for pointing out that it should be included. If the decision problem is uncertain (i.e., if $0 < \mathbb{P}(Y = 1 | X = x) < 1$ sometimes), then there are instances $x$ for which no decision maker could be sure about the correct decision $Y$, even when they have full knowledge of the joint distribution of $(X, Y)$ as assumed. This is what we are referring to as decisions under uncertainty—not a case where the decision maker is still learning about the distribution of $(X, Y)$.
>
>    Our definition of partial observability is somewhat nonstandard since we are using a statistical decision theory framework, IDT, instead of a (PO)MDP framework. In Appendix B, we show that IDT is equivalent to a POMDP; this section may provide some clarity on the connection to partial observability. In particular, we consider the unseen state to be the correct decision $Y$ and the observation to be $X$. The decision maker can use a particular $x$ to infer a posterior distribution over the correct decision Y (i.e., $\mathbb{P}(Y = 1 | X = x)$), and then must make a decision under this uncertainty.
>
>  * **MD smoothness:** this condition is a bit difficult to understand, but we attempted to give some more intuition for it in the appendices. It basically precludes situations where the optimal decision rule $h_c$ for cost parameter $c$ in one hypothesis classes $\mathcal{H}$ is also contained in a different hypothesis class $\mathcal{H}'$, but for a different cost parameter $c'$, the optimal decision rule $h_{c'} \in \mathcal{H}$ is not in the other hypothesis class $\mathcal{H}'$.  In Lemma E.2, we give an example of how MD-smoothness can be established for a particular family of hypothesis classes. The main requirement of the lemma is that the distribution of the induced posterior probability $q_{\mathcal{H}_S}(X)$ has a bounded density, which seems reasonable to assume for many real-world distributions. We also give some further intuition for the necessity of MD smoothness in Appendix D.2, where we provide an example of when a lack of MD smoothness prevents identifiability of the loss parameter c. Appendix D.2 is already referenced below the definition of MD smoothness; we will also include the above reference to Lemma E.2 there and the intuitive definition above.
>
>  * **Missing references**: thank you for the references to Mindermann et al. and Biyik et al. Below, we compare their results to ours; we will include this comparison in the final paper.
>
>    Similarly to us, Mindermann et al. explore a setting where the ambiguity about a person’s preferences can be reduced. In their case (AIRD), preferences are learned by actively querying the person for their choice from a set of suboptimal reward functions. Biyik et al. also investigate active learning to identify preferences. In contrast, our analysis concerns a purely passive, observational setting. In many realistic settings, it may be difficult or impossible to actively query a person about different reward functions or scenarios. Furthermore, active learning can lead to a description-experience gap (described in line 118 in our paper). In these cases, AIRD and other active learning methods cannot be applied. Thus, our work is complementary to these methods, analyzing when preferences can still be identified if only observational data is available.
>
>    Biyik et al. do observe that a particular type of uncertainty makes active queries more difficult for a person to answer. However, this is a different type of uncertainty from what we consider. Intuitively, Biyik et al. argue that uncertainty about the human makes preference learning harder, while we argue that uncertainty by the human makes it easier. In particular, the type of uncertainty Biyik et al. consider comes from the stochastic human model they consider, where a person randomly chooses among decisions with very similar rewards. Since they assume stochastic decision making, there is uncertainty about how a person will answer given a particular query or observation—this is the type of uncertainty they attempt to reduce. In contrast, we consider decision makers with deterministic decision rules: given the same observation, they always make the same decision. In our model, there may be uncertainty about the correct decision $Y$ given the observation $X$, but the human-made decision $\hat{Y}$ is deterministic given $X$. In other words, the person may be uncertain about the state of the world (e.g., if a person has a disease), but is certain about what decision to make (e.g., to quarantine them). Since the two definitions of uncertainty are different, our results are not in conflict with those of Biyik et al. This comparison highlights some interesting differences between our model of suboptimality (choosing a deterministic decision rule from a restricted hypothesis class) and that of Biyik et al. (Boltzmann rationality or noisy optimality). Further study of these differences could be a fruitful direction for future work.
>
>  * **Typo:** we will fix the typo. Thank you for pointing it out.

---

> > ### Comment · Reviewer_gz3s · 2021-08-11
> > **Re definition of uncertainty**
> >
> > 1)
> >
> > > Intuitively, Biyik et al. argue that uncertainty about the human makes preference learning harder, while we argue that uncertainty by the human makes it easier.
> >
> > In contrast to your result, Biyik/Mindermann also argue that uncertainty _by the human_ makes preference learning harder. From the abstract, they focus on "questions that naturally account for the human's ability to answer" - i.e. questions where the human is not uncertain. Formally, for an 'easy' question, the human policy p(y=1 | x, c) has low entropy (given access to the true parameters c that define the human's policy).
> >
> > If you are using a different notion of "uncertainty by the human" can you please clarify what this notion is?
> >
> > The difference might be that previous work shows that human uncertainty _for a given x_ is harmful for learning p(y | x) as well as parameters c, whereas your work shows that human uncertainty is still needed for _some_ x in order to identify c.
> >
> > 2)
> >
> > Furthermore, if "the [human] decision maker knows the true distribution of observations  and correct decisions (targets)", what is the human uncertain about? If it is the outcome Y of the human's own decision-making, we're back in the setting under 1) where previous work has shown that human uncertainty hurts preference learning. (As defined by the entropy of p(y=1 | x, c)).
> >
> > 3)
> >
> > > Given this framework, we define a decision problem with no uncertainty as one where [...]  almost surely.
> >
> > Please also define the probability distributions over which P goes. Not doing this makes the paper quite confusing as it is regularly unclear whether the random variable is X or c (or Y).

---

> > > ### Author Response · Authors · 2021-08-13
> > > **Clarifying our setting**
> > >
> > > Thank you for your response. We believe there may still be some confusion between the correct decision or label, $Y$, and the decision actually made by the human, $\hat{Y}$. The human’s policy is a decision rule $h$ which maps from observations $X$ to decisions $\hat{Y}$, i.e. $\hat{Y} = h(X)$. Since this decision rule is deterministic, the made decision $\hat{Y}$ is always deterministic given $X$. Thus the human’s policy has no entropy, i.e. $H(\hat{Y} \mid X) = 0$. However, the correct decision $Y$ may be uncertain given the observation $X$; *this* is the kind of uncertainty consider in the paper. For instance, a doctor may be deciding whether or not to operate on a patient with symptoms $x$ that he is 50% sure is having a heart attack. If the doctor knew for sure the patient was having a heart attack, he would always operate; if not, he would never operate. So $P(Y = 1 | X = x) = 0.5$, where $Y = 1$ means the correct decision is to operate. However, the doctor’s decision rule may be to operate on any patients with at least a 10% chance of having a heart attack, so $h(x) = 1$ and thus $P(\hat{Y} = 1 | X = x) = P(h(x) = 1 | X = x) = 1$. Thus, there is uncertainty about the *correct* decision, but not about the *made* decision. Below, we have addressed your specific concerns.
> > >
> > >
> > >  1. We do agree that Biyik et al. argue that it is more difficult for humans to make decisions under the type of uncertainty you mentioned. However, our analysis shows that it is necessary to observe these uncertain decisions in order to exactly identify a human’s preferences. If we do not observe decisions arbitrarily close to the person’s decision boundary, we cannot exactly characterize the decision boundary and thus we cannot recover the human’s exact preferences. Thus, there is a tradeoff between the ease of the decision problem for the human and the identifiability of their preferences. That is, uncertainty may make the human’s decision problem more difficult but our problem of identifying preferences easier. We will discuss this tradeoff in more detail in the final version.
> > >  2. As mentioned above, $Y$ is the *correct* decision while $\hat{Y}$ is the *made* decision. In our setting, $Y$ may be uncertain given $X$ but $\hat{Y}$ is deterministic given $X$ since $\hat{Y} = h(X)$ for some decision rule $h$. That is, there is no entropy in the human’s policy or decision rule.
> > >  3. We will add to all expectations and probabilities throughout the paper the precise distributions over which they are taken. Note that $c$ is not actually a random variable; rather, it is a fixed parameter of the IDT setting. With our definition in Section 3 of $\mathcal{D}$ as the joint distribution of $(X, Y)$, we can rewrite our definition of a decision problem with no uncertainty as one where $P_{(X, Y) \sim \mathcal{D}}(Y = 1 | X = x) \in \\{0, 1\\}$ almost surely. In other words, the correct answer is always deterministic given the observation. Note that this definition does not depend on the cost parameter $c$, the decision rule $h$, or the made decision $\hat{Y}$.

---

### Official Review · Reviewer_FKEy · 2021-07-20

**Rating:** 7
**Confidence:** 4

**Summary:**

The paper studies inverse decision theory---the problem of identifying the loss function of given optimal (under a given hypothesis class) state-action (covariates-label) pairs in a binary classification setting. The contributions of the paper are two-fold: the authors provide (1) a nice formulation of inverse learning problems where one can analytically study suboptimal behaviors and what makes inverse learning easier/harder; (2) identifiability and sample complexity results for identifying the loss function when the hypothesis class is either realizable or unrealizable.

**Limitations And Societal Impact:**

The authors have adequately addressed the limitations and potential negative societal impact of their work.

**Main Review:**

Though most inverse learning paper focuses on sequential decision-making settings (e.g., inverse reinforcement learning), it is great to see an inverse supervised learning paper that has a more distilled analysis of inverse learning using suboptimal behavior. By considering cases where suboptimality comes into play due to the unrealizable hypothesis class, the authors provide an interesting way of defining suboptimality in inverse learning settings (which I believe is novel). Here are some questions/comments:

(1) Is the assumption on P(q_H(X) \in (c, c+ \rho)) \geq p_c \rho and  P(q_H(X) \in (c-\epsilon, c)) \geq p_c \epsilon necessary? Some experimental results on how the estimator performs under cases when such assumptions don't hold would help the readers understand this assumption better.
(2) Will this intuition that identification is easier when the problem is more uncertain extends to cases where the loss functions are not 0/1 loss but some surrogate losses (e.g., logistic loss)?
(3) As supervised learning can be viewed as an RL instance, could the results in this paper suggest some lower bound on identifiability of IRL under suboptimal behaviors?

**Time Spent Reviewing:**

2 hours

---

> ### Author Response · Authors · 2021-08-10
> **Response to Reviewer FKEy**
>
> Thank you for your positive review and insightful feedback. Here are responses to each of the questions and comments you made:
>
>  1. The assumption on the distribution of $q_\mathcal{H}(X)$ is necessary. Theorem 4.11 shows that the sample complexity of identifying the cost parameter is in fact lower bounded proportionally to $1/p_c$, so as $p_c \to 0$ the sample complexity goes to $\infty$. As some intuition, imagine that this assumption is not satisfied; say that $\mathbb{P}(q_\mathcal{H}(X) \in (c, c+\epsilon)) = 0$. Consider two optimal decision rules in $\mathcal{H}$ for cost parameters $c$ and $c+\epsilon$, respectively. According to Lemma 4.1/Corollary 4.6, these decision rules only differ when $q_\mathcal{H}(X) \in (c, c+\epsilon)$. Since the assumption is not satisfied, we will never observe such a decision. Thus, it is impossible to tell apart the two optimal decision rules with different cost parameters. We will include this explanation in the main text as well as a figure illustrating why this assumption is necessary for identifiability.
>  2. The intuition that preferences are more identifiable for uncertain decisions does extend to surrogate losses. Rosasco et al. [1] show that for most reasonable surrogate losses (hinge, logistic, etc.), the optimal classifier under the surrogate loss is equivalent to the Bayes optimal classifier for the 0-1 loss (see section 4.2 of their paper). Thus, for an optimal decision maker using a surrogate loss, the analysis from our Section 4.1 applies; preferences are identifiable only for uncertain decision problems.
>
>     In the case of a suboptimal decision maker with restricted hypothesis class, the optimal decision rule for a surrogate loss may not be the same as the optimal rule for the 0-1 loss. However, the optimal rules will agree in the case that there is no uncertainty (the realizable case), so this is sufficient to show that the cost parameter is not identifiable if there is no uncertainty in the decision problem.
>
>     We will add this analysis of surrogate loss functions to the appendix and analyze under which surrogate losses our analysis for the 0-1 loss is applicable.
>
>  3. We believe that our results can be extended to preference learning in more general sequential decision-making cases like IRL. However, since sequential decision-making presents many additional complexities compared to single decisions, we leave such analysis to future work. As we note in line 111, “We begin with the restricted setting of IDT but hope to extend to sequential decision making in the future.”
>
>
> [1] Rosasco, De Vito, Caponnetto, Piana, and Verri. Are Loss Functions All the Same?. Neural Computation.

---

> > ### Comment · Reviewer_FKEy · 2021-08-18
> > **Response to rebuttal**
> >
> > Thanks for the detailed reply. I am satisfied with the responses to comments #1, #2, and #3.

---

### Decision · Program_Chairs · 2021-09-27

**Decision:**

Accept (Spotlight)

**Comment:**

This paper analyzes the inverse decision theory task of recovering the loss function of a decision maker making (observational) decisions under uncertainty. The paper's surprising insight is that uncertainty of the decision maker can enable better loss function recovery. This is supported with sample complexity bounds. There were some initial concerns among the reviewers about clarity in distinguishing "clear" vs. "uncertain" decisions and the positioning of the work among some other papers, but the author(s) was/were able to alleviate those concerns and the reviewers are satisfied with the intended direction of the final version of the paper. The reviewers discussed the lack of experiments in the paper, concluding that experiments demonstrating the analyzed benefits---particularly with suboptimal decision makers---would enhance the paper, but that the paper was a significant contribution without additional experiments and worthy of acceptance.